# DISTRIBUTIONAL SOBOLEV REINFORCEMENT LEARNING

## ABSTRACT

Distributional reinforcement learning (DRL) is a framework for learning a complete distribution over returns, rather than merely estimating expectations. In this paper, we extend DRL on continuous state-action spaces by modeling not only the distribution over the scalar state-action value function but also its gradient. We refer to this method as *Distributional Sobolev training*. Inspired by Stochastic Value Gradients (SVG), we achieve this by leveraging a one-step world model of the reward and transition distributions implemented using a conditional Variational Autoencoder (cVAE). Our approach is sample-based and relies on Maximum Mean Discrepancy (MMD) to instantiate the distributional Bellman operator. We first showcase the method on a toy supervised learning problem. We then validate our algorithm in several Mujoco/Brax environments.

## 1 INTRODUCTION

Reinforcement learning tackles sequential decision-making, where an agent maximizes cumulative rewards from the environment. In recent years deep reinforcement learning (DRL) has achieved remarkable success, exemplified by the Deep Q-Network (DQN) (Mnih et al., 2015), which reached human-level performance in Atari games. Similarly, significant progress has been made in continuous control tasks, where policies are parameterized by neural networks and optimized using gradient ascent. In the model-free setting, two families of methods exist for computing policy gradients. The first samples returns from the environment and relies on likelihood ratio estimators (Sutton et al., 1999; Williams, 1992). The second approach, commonly referred to as value-based, computes the gradient of a learned state-action value function via backpropagation and uses it as the policy gradient (Lillicrap et al., 2016; Fujimoto et al., 2018; Haarnoja et al., 2018). This work focuses on improving the latter approach.

In value-based methods, the policy gradient is derived from a learned critic, meaning any improvement in value function training could enhance policy optimization. In this paper, we propose a unified framework that integrates two orthogonal but complementary improvements. First, we incorporate gradient information in the training of the value function (Fairbank, 2008; D'Oro & Jaskowski, 2020; Czarnecki et al., 2017). Second, we borrow ideas from distributional reinforcement learning Bellemare et al. (2017) and model uncertainty not only over returns but also over *action-gradients*. This allows us to capture intrinsic environmental uncertainty more acurately by leveraging more information from the observations collected on the environment. Since this new framework models a distribution over both the output and input gradients of the critic, we refer to it as **Distributional Sobolev Reinforcement Learning**.

Value functions are typically trained using temporal difference learning (Sutton, 1988), where targets are based on observed environment transitions. The policy is improved by backpropagating action-gradients through the policy network. However, as noted by D'Oro & Jaskowski (2020), action-gradients learned via temporal difference rely on smoothness assumptions on the true value function (Lillicrap et al., 2016). Similar to D'Oro & Jaskowski (2020), we incorporate gradient information into value function training by leveraging a learned model of transition dynamics and rewards, providing a differentiable proxy for the environment (Heess et al., 2015). Thus our world model is not used for imagining new samples as is common in model-based reinforcement learning (MBRL) (Sutton, 1991; D'Oro & Jaskowski, 2020)

Moreover, many environments exhibit irreducible uncertainty in transitions and rewards. Distributional RL (Bellemare et al., 2017) models this uncertainty as a distribution over returns rather than focusing on expected return, leading to empirical improvements in various tasks (Dabney et al., 2018a; Barth-Maron et al., 2018; Hessel et al., 2017). We argue that the stochastic nature of return should reflect on their *action-gradient* and even more so for tasks involving large action spaces. Hence, we extend distributional modeling over both returns and their gradients, motivated by the sample efficiency gains observed in prior work (Hessel et al., 2017; Dabney et al., 2018b; Barth-Maron et al., 2018; Dabney et al., 2018a). As we will discuss, this combined framework not only improves sample efficiency but also exhibits properties that could benefit broader machine learning fields.

**Paper contributions**  By integrating gradient-based training with uncertainty modeling, we aim to enhance both policy and value function learning. Doing so will necessitate to model random variables as well as their gradient. This will require a flexible generative model whose output and input-gradient can be differentiated. Hence, one contribution of this work is to introduce **distributional Sobolev training** and propose a way to implement this new paradigm. As most reinforcement learning environments are not differentiable, we will rely on inference using a conditional variational autoencoder (cVAE) Sohn et al. (2015) to predict gradients **from observed samples** (in contrast to doing it via imagination). This is another contribution of this paper as we put stochastic value gradients (SVG) Heess et al. (2015) to a new use (distributional Sobolev training) and doing so with a more expressive class of neural network (cVAEs).

**Paper structure**  The remainder of the paper is structured as follows: Section 2 covers the key concepts and notations related to deterministic policy gradients, neural network training with gradient information, and distributional reinforcement learning. In Section 3, we detail our proposed method and algorithm. Section 4 presents empirical results, showcasing experiments on both toy examples and real-world tasks.

## 2 BACKGROUND

### 2.1 REINFORCEMENT LEARNING

In this work, we address a reinforcement learning problem where an agent interacts with the environment to maximize cumulative rewards. The agent operates in continuous state $\mathcal{S}$ and action $\mathcal{A}$ spaces, with transitions governed by a distribution $P : \mathcal{S} \times \mathcal{A} \to \mathcal{P}(\mathcal{S})$ and rewards modeled by $R : \mathcal{S} \times \mathcal{A} \to \mathcal{P}(\mathbb{R})$. The initial state distribution is $\mu \in \mathcal{P}(\mathcal{S})$. The deterministic policy $\pi_\theta$, parameterized by $\theta$, maps states to actions. The $\gamma$-discounted state occupancy measure is given by $d_\mu^{\pi_\theta} = (1-\gamma) \sum_{t=0}^{\infty} \gamma^t \Pr(s'|\pi_\theta, \mu)$, as derived in D'Oro & Jaskowski (2020) and Silver et al. (2014). The state-action value function, $Q^\pi(s, a)$, defines the expected future rewards starting from state $s$ and action $a$, i.e., $Q^\pi(s, a) = \mathbb{E}\left[\sum_{t=0}^{\infty} \gamma^t r(s_t, a_t) \mid s_0 = s, a_0 = a\right]$. Finally, the reinforcement learning objective is defined as

$$J(\theta) = \mathbb{E}_{s \sim \mu}\left[Q^{\pi_\theta}(s, \pi_\theta(s))\right]. \tag{1}$$

The deterministic policy gradient theorem (Silver et al., 2014) states that *under some mild regularity conditions on the Markov Decision Process (MDP)*, the gradient of the RL objective is

$$\nabla_\theta J(\theta) = \frac{1}{1-\gamma} \mathbb{E}_{s \sim d_\mu^{\pi_\theta}}\left[\nabla_\theta \pi_\theta(s) \left. \nabla_a Q^{\pi_\theta}(s, a)\right|_{a=\pi_\theta(s)}\right]. \tag{2}$$

Equation 2 assumes access to the true Q-function of the policy. However, it can be approximated by a learned critic $Q_\phi$ with parameters $\phi$, using temporal difference after introducing the Bellman operator

$$(\mathcal{T}_\pi Q)(s, a) = \mathbb{E}\left[R(s, a)\right] + \gamma \mathbb{E}\left[Q(s', \pi(s')) \mid s, a\right]. \tag{3}$$

Most of the time this leads to a regression loss where the bootstrapped target is estimated using a delayed target policy and value network with parameters $\theta', \phi'$. Thanks to the off-policyness of this scheme, the expectation is evaluated under the distribution from a replay buffer denoted $B$ (Lillicrap et al., 2016; Fujimoto et al., 2018; Haarnoja et al., 2018). We first define the bootstrapped target $\delta^\pi(s, a, s') = r + \gamma Q_{\phi'}(s', \pi_{\theta'}(s'))$ and use distance $d(.|.)$ to evaluate the critic's loss

$$\mathcal{L}_Q(\phi) = \mathbb{E}_{(s,a,r,s') \sim B}\left[d\left(Q_\phi(s, a)|\delta\right)\right]. \tag{4}$$

## 2.2 Distributional reinforcement learning (DRL)

Distributional reinforcement learning was proposed in Bellemare et al. (2017) and extends equation 3 by considering the full distribution over returns instead of the statistical mean. We denote the state-action value random variable $Z^\pi(s, a)$, which follows the distribution $\eta^\pi(s, a)$. Firstly, the expected state-action value function is related to the distributional value function as $Q^\pi(s, a) = \mathbb{E}[Z^\pi(s, a)]$. Following notation from Zhang et al. (2021); Rowland et al. (2019) we define a Bellman operator over the distribution of random return

$$(\mathcal{T}_\pi^D \eta)(s, a) = \int_S \int_A (f_{r,\gamma})_\# \eta(s', a') \pi(da' \mid s') P(ds' \mid s, a), \tag{5}$$

where $f_{r,\gamma}(x) = r + \gamma x$ and $f \# \eta$ is the pushforward measure as defined in Rowland et al. (2018).

We can also define a Bellman operator more similarly to equation 3, but this time equality is in the probability law of the random variables (Bellemare et al., 2017).

$$(\mathcal{T}_\pi^D) Z(s, a) \stackrel{D}{=} R(s, a) + \gamma Z(s', \pi(s')) \quad \text{where} \quad s' \sim P(\cdot \mid s, a). \tag{6}$$

As for the non-distributional case, the distributional critic will be parametrized by a neural network. A loss can be derived from the distributional Bellman operator which this time will have to work on distributions. Numerous parametrizations of this one-dimensional distribution have been proposed based on quantiles (Dabney et al., 2018b;a; Singh et al., 2022; Yue et al., 2020), discrete categorical (Bellemare et al., 2017; Barth-Maron et al., 2018) or sample/pseudo-sample based (Freirich et al., 2019; Zhang et al., 2021; Doan et al., 2018; Nguyen et al., 2020).

From these parameterized distributions a statistical loss is minimized which we canonically denote as

$$\mathcal{L}^D = \mathbb{E}_{s \sim d_\mu^{\pi_\theta}} \left[ d((\mathcal{T}_\pi^D) Z(s, a) | Z(s, a)) \right] \quad \text{where} \quad a = \pi(s). \tag{7}$$

From previous work (Barth-Maron et al., 2018) the deterministic policy gradient (Eq 2) can be extended to the distributional case simply by plugging $Q^\pi(s, a) = \mathbb{E}[Z^\pi(s, a)]$

$$\nabla_\theta J(\theta) = \frac{1}{1 - \gamma} \mathbb{E}_{s \sim d_\mu^{\pi_\theta}} \left[ \nabla_\theta \pi_\theta(s) \mathbb{E}[\nabla_a Z^{\pi_\theta}(s, a)]|_{a = \pi_\theta(s)} \right], \tag{8}$$

where $\nabla_a Z_\theta^\pi(s, a)$ is a random variable that can be interpreted as the action-gradient of realizations of the random variable $z \sim Z(s, a)$. We detail this intuitive interpretation in Appendix A.1.

In this work we will sometimes make use of a modification on how the temporal difference is estimated. We refer to this modified setting as *N-step return*. It can be seen as modifying the Bellman operator as

$$\left(T_\pi^{D,N}\right) Z(s_0, a_0) \stackrel{D}{=} \sum_{n=0}^{N-1} \gamma^n r(s_n, a_n) + \gamma^N Z(s_N, \pi(s_N)) \quad s_{i+1} \sim P(\cdot \mid s_i, a_i). \tag{9}$$

In practice, the actions in the N-step returns of Eq. 9, are drawn from an exploration policy which for deterministic policies is often $\pi_{\exp}(s) = \pi(s) + \epsilon$ where $\epsilon$ is some exploration noise. This trick is most often used to improve sample efficiency (Hessel et al., 2017). However, as the target now depends on some action noise, it also has the nice property of making deterministic environments stochastic from the perspective of policy evaluation.

## 2.3 Sobolev training and Value gradients

Sobolev training of neural networks (Czarnecki et al., 2017) suggests using derivatives information, when available, to train neural networks. Since the derivatives of a neural network are differentiable, a loss function can be constructed and minimized using stochastic optimization. Given a target differentiable function $F : \mathbb{R}^a \to \mathbb{R}^b$, we train a neural network $f_\varphi$ with learnable parameters $\varphi$, using a loss function incorporating both the output and its derivatives:

$$\mathcal{L}^S(\varphi; x) = \|F(x) - f_\varphi(x)\|^2 + \lambda_S^1 \|\nabla_x F(x) - \nabla_x f_\varphi(x)\|^2. \tag{10}$$

Strong empirical evidence from Czarnecki et al. (2017); D'Oro & Jaskowski (2020) indicates that modeling the gradient of a function using the gradient of a neural network, trained along with zero-order information, results in both greater accuracy and stability. We refer to this observation as the **Sobolev inductive bias**.

## 3 APPROACH

### 3.1 LEARNING A USEFUL CRITIC

In value-based methods such as Lillicrap et al. (2016); Fujimoto et al. (2018); Haarnoja et al. (2018), the actor's training signal relies solely on the learned critic, meaning that "actor can only be as good as allowed by its critic" (D'Oro & Jaskowski, 2020). However, these learned critics are inherently imperfect, partly due to the mean predictions that cannot capture the underlying uncertainty in returns. Distributional RL, as discussed in Section 2.2, addresses this by modeling the return distribution. Both involved temporal-difference learning on the returns via Eq. 4 or distribution of returns bia Eq. 7. Another fundamental issues is that minimizing either expectation or distributional TD-error does but an additional improvement can be made by directly considering the action-gradient of the critic in the training objective.

**Proposition 3.1** *Let $\pi$ be an $L_\pi$-Lipschitz continuous policy, and suppose $G(s)$ and $\hat{G}(s)$ are the true and estimated distributions of the action gradients $\nabla_a Z^\pi(s, a)$ and $\nabla_a \hat{Z}(s, a)$ at $a = \pi(s)$, respectively. The Wasserstein-1 distance $\mathcal{W}_1$ between $G(s)$ and $\hat{G}(s)$ is defined as:*

$$\mathcal{W}_1(G(s), \hat{G}(s)) = \inf_{\gamma \in \Pi(G(s), \hat{G}(s))} \mathbb{E}_{(X,Y) \sim \gamma} \left[ \| X - Y \| \right], \tag{11}$$

*where $\Pi(G(s), \hat{G}(s))$ is the set of all couplings of $G(s)$ and $\hat{G}(s)$. Then, the error between the true policy gradient $\nabla_\theta J(\theta)$ and estimated policy gradients $\nabla_\theta \hat{J}(\theta)$ that uses the expectation of $\hat{G(s)}$*

$$\left\| \nabla_\theta J(\theta) - \nabla_\theta \hat{J}(\theta) \right\| \leq \frac{L_\pi}{1 - \gamma} \mathbb{E}_{s \sim d_\pi^\mu} \left[ \mathcal{W}_1 \left( \nabla_a Z^\pi(s, \pi(s)), \nabla_a \hat{Z}(s, \pi(s)) \right) \right]. \tag{12}$$

This proposition (proof in Appendix A.2) is a distributional generalization of Proposition 3.1 from D'Oro & Jaskowski (2020). The Lipschitz assumption for $\pi$ is commonly satisfied using neural networks for approximation.

Similarly to D'Oro & Jaskowski (2020) we can induce an optimization problem for the critic from Proposition 3.1. Indeed, it suggests we can approximate the true policy gradient by matching the action-gradient in the distributional sense. As is common with temporal difference, the true distribution will be approximated using *bootstrapping* which gives the following optimization problem.

$$\hat{Z} \in \arg\min_{\hat{Z} \in \mathcal{Z}} \mathbb{E}_{\substack{s \sim d_\mu^\pi(s) \\ (s', r) \sim p(s', r | s, \pi_\theta(s))}} \left[ \mathcal{W}_1 \left( \nabla_a \hat{Z}(s, a), \nabla_a r(s, a) + \gamma \nabla_a \hat{Z}(s', \pi_\theta(s')) \right) \right]. \tag{13}$$

We note that Eq. 13 assumes to have a known and differentiable dynamics $p$. We maintain this assumption for the time being and will relax it a subsequent section. In the next section we formalize the notions necessary to instantiate a working implementation of this optimization problem.

### 3.2 DISTRIBUTIONAL SOBOLEV TRAINING

In order to instantiate Eq. 13 while benefiting from Sobolev inductive bias introduced in Section 2.3, we need to extend distributional RL to model the joint distribution of the action-gradient alongside the standard scalar return. This first requires defining notions similar to expected return and random return. Let's denote the first order **random action Sobolev return** $Z^{S_a}(s, a)$ that is the joint random variable formed by the concatenation of random return and the action-gradient of random return.

$$Z^{S_a}(s, a) = \left[ \sum_{t=0}^\infty \gamma^t r(s_t, a_t); \nabla_a \left( \sum_{t=0}^\infty \gamma^t r(s_t, a_t) \right) \right] \quad \text{where} \quad s_0 = s, \ a_0 = a. \tag{14}$$

Here, $\nabla_a$ denotes the gradient taken with respect to the action variable $a$, which is indexed by $a_0 = a$. Similarly, as presented in Section. 2.2 from the random variable we can define an expectation as the **expected action Sobolev return** $Q^{S_a}(s, a) = \mathbb{E}\left[ Z^{S_a}(s, a) \right]$. The definition from Eq. 14 implies to model a random variable of size $|\mathcal{A}| + 1$ and suggests to define a Bellman operator. We

borrow notation from Zhang et al. (2021); Rowland et al. (2019) to define such a Bellman operator on the multivariate random Sobolev return.

$$(\mathcal{T}_\pi^{S_a} \eta^{S_a})(s,a) \overset{D}{=} \int_S \int_A \int_\mathbb{R} (\mathbf{f}_{s,a,r,s',\gamma}^S)_\# \eta^{S_a}(s',a') \, R(dr \mid s,a) \, \pi(da' \mid s') \, P(ds' \mid s,a), \quad (15)$$

where $\eta^{S_a}(s,a)$ is the joint distribution of the random variable $Z^{S_a}(s,a)$. The transformation $\mathbf{f}_{s,a,r,s',\gamma}^{S_a} : |\mathcal{A}| + 1 \to |\mathcal{A}| + 1$ describes how to map an action Sobolev return from the next state-action pair $(s',a')$ into an action Sobolev return at the current state-action pair $(s,a)$. We denote $x = [x^{\text{return}}; x^{\text{action}}]$ as the joint random variable over return and action-gradient. The transformation works as follows

$$\mathbf{f}_{s,a,r,s',\gamma}^{S_a}(x) = \begin{bmatrix} f_{s,a,r,s',\gamma}^{\text{return}}(x) \\ f_{s,a,r,s',\gamma}^{\text{action}}(x) \end{bmatrix}, \quad (16)$$

where

$$f_{s,a,r,s',\gamma}^{\text{return}}(x) = r + \gamma x^{\text{return}}, \quad (17)$$

and

$$f_{s,a,r,s',\gamma}^{\text{action}}(x) = \frac{\partial}{\partial a} r(s,a) + \gamma \frac{\partial s'}{\partial a} \left( \frac{\partial}{\partial s'} x^{\text{return}} + \frac{\partial a'}{\partial s'} x^{\text{action}} \right). \quad (18)$$

The usual operator from Eq. 17 simply acts on the random return in the next state-action pair $(s', \pi(s'))$. Thus, under operator of Eq. 16 the first dimension undergoes the conventional transformation as described in Section 2.2. The second part of the operator in Eq. 18 is novel as it acts on the action gradient of the random return. It takes as an input the random return $x^{\text{return}}$ and random action-gradient $x^{\text{action}}$ and depends on the reward and dynamics distributions through samples $(s', r)$. Essentially, this can be seen as taking a sample from the distribution induced by operator Eq. 17 and taking its action derivative. The proof for Eq. 18 is provided in Appendix A.3. Notably, assuming the dynamics are know and differentiable, there is no need to manually implement equation 18, as it can be implicitly computed using automatic differentiation (Baydin et al., 2018; Paszke et al., 2019; Bradbury et al., 2018).

### 3.3 TOWARDS A SURROGATE FOR SOBOLEV BELLMAN OPERATOR

In this section, we outline the requirements and rationale behind our choice of generative model. We then describe the final model and its training procedure. Our approach is primarily driven by a collection of practical considerations, as detailed below.

**Distributional Sobolev critic** We preserve the *Sobolev inductive bias* introduced in Section 2.3, necessitating a generative model where both the output and its input-gradient are treated as random variables. While the reparameterization trick (Kingma, 2013) could be used with a conditional Gaussian distribution, determining how to distribute the gradient of the samples with respect to the conditioners is less straightforward. To address this, we employ a sample-based approach that circumvents likelihood estimation and relies solely on sampled data.

Moreover, we found that both discrete categorical representations (Bellemare et al., 2017; Barth-Maron et al., 2018) and quantile-based representations (Dabney et al., 2018b;a) do not scale tractably to higher dimensions. These considerations further motivated us to adopt a sample-based approach for the distributional Sobolev critic, similar in spirit to Singh et al. (2022); Freirich et al. (2019). As a result, the distributional critic is structured as a generative model that deterministically maps noise to samples.

**Approximate Bellman operator via MMD minimization** Similarly to Eq. 3, the Bellman operator introduced in Eq. 15 requires defining a notion of distance between distributions. However, the integral in Eq. 15 is intractable, and we only have access to sampled transitions. Therefore, it is essential to select an objective that can be optimized using stochastic methods to effectively approximate the Bellman operator.

Because of its simplicity we chose to minimize the Maximum Mean Discrepancy (MMD) between the Sobolev returns $\eta^{S_a}$ and bootstrapped Sobolev returns distributions $T_\pi^{S_a} \eta^{S_a}$

$$\text{MMD}^2(P, Q; k) = \mathbb{E}_{x,x' \sim P}[k(x,x')] - 2\mathbb{E}_{x \sim P, y \sim Q}[k(x,y)] + \mathbb{E}_{y,y' \sim Q}[k(y,y')], \quad (19)$$

where $X$ and $X'$ are independent random variables sampled from the distribution $P$, and $Y$ and $Y'$ are independent random variables sampled from the distribution $Q$.

The kernel function $k(x, y)$ serves as a measure of similarity between two inputs $x$ and $y$. It can take various forms, with common choice being the Gaussian radial basis function (RBF) kernel:

$$k_\sigma^{\text{RBF}}(x, y) = \exp\left(-\frac{\|x - y\|^2}{2\sigma^2}\right). \tag{20}$$

As introduced in Gretton et al. (2012), we can get an unbiased estimate of the MMD by computing pairwise similarities between the samples using the kernel function $k(x, y)$ as

$$\widehat{\text{MMD}}_u^2(\{x_i\}, \{y_i\}; k) := \frac{1}{m(m-1)} \sum_{i \neq j} k(x_i, x_j) + k(y_i, y_j) - 2k(x_i, y_j). \tag{21}$$

However, it is more common for the following biased estimator to be used in the context of distributional reinforcement learning, mostly because of its claimed lower variance (Nguyen et al., 2020)

$$\widehat{\text{MMD}}_b^2(\{x_i\}, \{y_i\}; k) := \frac{1}{m^2} \sum_{i,j} k(x_i, x_j) + k(y_i, y_j) - 2k(x_i, y_j). \tag{22}$$

In essence, our distributional critic can be viewed as a **conditional Generative Moment Matching Network (cGMMN)** (Li et al., 2015; Bińkowski et al., 2021; Oskarsson, 2020). The Maximum Mean discrepancy has already been considered in various distributional RL algorithms Nguyen et al. (2020); Killingberg & Langseth (2023); Zhang et al. (2021) where the random variable is always modeled through pseudo-samples, represented by multiple fixed outputs from the critic.

**Convergence and kernel choice**  Previous works have shown that the kernel choice was of paramount importance in order to effectively use MMD in Distributional Reinforcement Learning (Nguyen et al., 2020; Killingberg & Langseth, 2023). The primary concern is whether the distributional Bellman operator is a contraction in terms of the distance between distribution $d$ introduced in Eq. 7 such that, following Killingberg & Langseth (2023), for some $k \in (0, 1)$ we have

$$d\left(\mathcal{T}_\pi^S Z_1^S, T_\pi^S Z_2^S\right) \leq k d\left(Z_1^S, Z_2^S\right) \tag{23}$$

If this condition holds, finding an estimator for $d$ can provide a principled way to design a loss function. Nguyen et al. (2020) demonstrated that $\mathcal{T}^D$ from Eq. 5 and 6 is a contraction in Maximum Mean Discrepancy (MMD) for specific kernels. Sufficient conditions on the kernel for the Bellman operator to be a contraction in MMD were further explored by Killingberg & Langseth (2023), who proposed the multiquadratic kernel $k_h^{\text{MQ}}(x, y) = -\sqrt{1 + h^2\|x - y\|_2^2}$. They demonstrated that this kernel exhibits beneficial properties and provides an empirical advantage over the commonly used RBF kernel.

**One-step world model**  To derive a computational scheme from Equation 15, we introduce a trained stochastic and differentiable world model $f(s, a) \to (\hat{s}', \hat{r})$, inspired by SVG(1) and MAGE (Heess et al., 2015; D'Oro & Jaskowski, 2020). This model mimics the environment and captures the inherent uncertainty in transition dynamics and rewards. Since we lack explicit distributions for these random variables, we rely on a sample-based method.

Assuming a stochastic environment represented by the function $g(s, a, \varepsilon) \to (s', r)$, where $\varepsilon \sim \rho_w(\varepsilon)$ is a random variable drawn from a distribution $\rho_w$, we can express the distributional Bellman equation over Sobolev returns as:

$$Z^S(s, a) \stackrel{D}{=} \mathbf{f}_{r,s',\gamma}^S\left(Z^S(s', a')\right) \quad \text{where} \quad (s', r) = g(s, a, \varepsilon),\ \varepsilon \sim \rho_w(\varepsilon),\ a' = \pi(s'). \tag{24}$$

We train a model that is both capable of inferring from observations and sampling new ones such that they can be used in lieu of the true environment in Eq. 24. This model is a conditional VAE (cVAE) (Sohn et al., 2015), where the encoder $q_\zeta(\varepsilon \mid s', r; s, a)$ maps the next state and reward, conditioned on the state-action pair $(s, a)$, into the latent space $\varepsilon$. The decoder $p_\psi(s', r \mid \varepsilon; s, a)$ then reconstructs the next state $s'$ and reward $r$ from the latent variable $\varepsilon$, conditioned on the state-action pair $(s, a)$. Assuming the cVAE is able to model the true conditional distribution while keeping the posterior close to the prior, then gradient information can be inferred from the reconstructed samples

as further discussed in Appendix A.5. For completeness, we provide a short introduction to cVAEs in Appendix A.4 and a small visualization of what inferring gradient from reconstructed samples involves in Appendix A.6

The world model is trained alongside both policy evaluation and policy improvement. By integrating the distributional critic and the world model into DDPG (Lillicrap et al., 2016), we introduce the **Distributional Sobolev Deterministic Policy Gradient (DSDPG)** algorithm. The procedure to estimate the update direction of the distributional critic is outlined in Algorithm 1, while an overview of the full DSDPG method is illustrated in Figure 1.

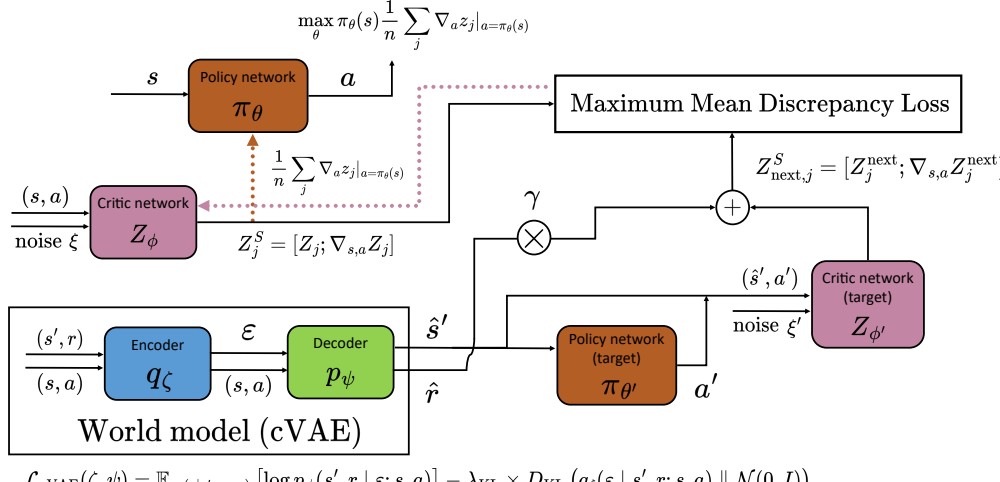

$$\mathcal{L}_{\text{cVAE}}(\zeta, \psi) = \mathbb{E}_{q_\zeta(\varepsilon|s',r;s,a)} \left[ \log p_\psi(s', r \mid \varepsilon; s, a) \right] - \lambda_{\text{KL}} \times D_{\text{KL}} \left( q_\zeta(\varepsilon \mid s', r; s, a) \parallel \mathcal{N}(0, I) \right)$$

Figure 1: Diagram of our Distributional Sobolev Deterministic Policy Gradient (DSDPG) algorithm. The distributional critic $Z_\phi$ (in pink) maps noise $\xi$ and a given state-action pair $(s, a)$ to samples from the distribution over Sobolev returns $Z^S$. The target samples are computed based on reconstructed samples $(\hat{s}', \hat{r})$ (lower branch) from a conditional Variational Auto-Encoder (cVAE) (in blue and green) acting as world model. The critic's output is then differentiated with respect to $a$ or $(s, a)$. The target and predicted distributions are compared using Maximum Mean Discrepancy (MMD). The policy network (in brown) is updated based on the empirical mean of the samples produced by the distributional critic (top of the figure). Gradient flows for the critic and the policy are shown in dashed lines. Diagram inspired by Singh et al. (2022).

## 4 RESULTS

### 4.1 TOY SUPERVISED LEARNING

To motivate our algorithm, we demonstrate its ability to learn the joint distribution over both the output and gradient of a random function in a supervised setup. Specifically, we show how incorporating gradient information enhances the modeling of such distributions.

The task involves a one-dimensional conditional distribution $p(y|x)$, which is a mixture of sinusoids sampled from the interval $[0, 5]$, with amplitude uncertainty over five discrete modes. We define $f(x; a) = a \times \sin(x)$, where the latent variable $a$ is uniformly drawn from $\{0, 1, 2, 3, 4\}$.

Figure 3 compares the Conditional Generative Moment Matching Network (cGMMN) and a regression-based method, both trained using stochastic gradient descent with identical architectures. Both models were trained in an unlimited data regime, where new pairs of $x$ and $a$ were drawn for each batch. In line with the reinforcement learning setup, for each $x$, four $a$ values were drawn with replacement, producing samples $(x, y_{1:4})$. More details about the empirical setup are provided in Appendix A.7. As expected, the cGMMN learns both the output and gradient distributions, while the regression model converges to the conditional expectation $E_a[f(x; a)]$.

Figure 3a shows MMD scores on the joint random variable $[f(x; a); \nabla_x f(x; a)]$, using a different kernel for evaluation. Figure 3b reports the L2 discrepancy between the regression model and the

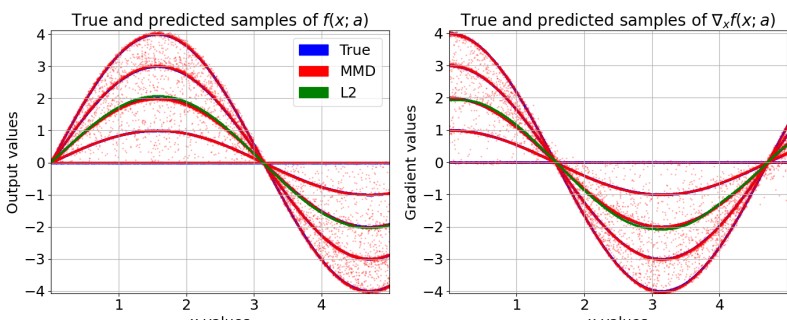

Figure 2: Supervised learning task on a toy problem featuring a sinusoidal function with 5 distinct modes, where the uncertainty is in the amplitude. The plot compares true samples from the random function (blue) against predictions from the cGMMN, trained using a distributional Sobolev approach with a mixture of RBF kernels (red), and a standard regression model trained with Sobolev L2 loss (green). The left panel shows samples from the output distribution, while the right panel presents samples from the gradient distribution.

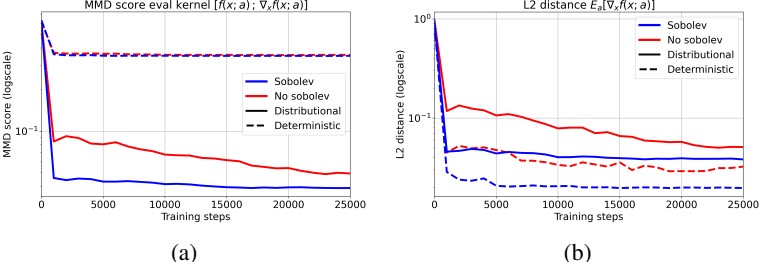

(a)                                             (b)

Figure 3: Evaluation metrics on toy supervised learning problem of distributional and deterministic methods with and without Sobolev training. Sobolev training uses gradient information during training. Distributional method is implemented as a conditional Generative Moment Matching Network (cGMMN). Deterministic (dashed) simply uses L2 regression. (Left) is average MMD score using an evaluation kernel different from the one used for training the generative moment matching network. (Right) average L2 loss on the predicted gradient versus the true gradient. Metrics averaged over 5 seeds.

---

**Algorithm 1** Gradient estimation of MMD$^2$ loss via reconstruction of transition samples

---

**Require:** Number of samples $M$, kernel $k$, discount factor $\gamma \in (0, 1)$
**Require:** Distributional critic $Z_\phi(s, a, \varepsilon)$
**Require:** Policy network $\pi_\theta(s)$
**Require:** Conditional VAE (cVAE) with encoder $q_\zeta(\varepsilon \mid s, a)$ and decoder $p_\psi(s', r \mid s, a, \varepsilon)$
    **Input:** Transition sample $(s, a, r, s')$
    **Input:** Online critic parameter $\phi$, target critic parameter $\phi'$
    **Input:** Target policy parameter $\theta'$
    **Output:** Gradient estimation of MMD with respect to $\phi$
  1: Encode $\varepsilon \sim q_\zeta(\varepsilon \mid s, a)$                    ▷ Encode latent variable based on $s$ and $a$
  2: Block the gradient on $\varepsilon$          ▷ No gradient backpropagation through the latent variable
  3: Reconstruct $\hat{s}'$ and $\hat{r}$ from the decoder $(\hat{s}', \hat{r}) \sim p_\psi(s', r \mid s, a, \varepsilon)$
  4: $a' \leftarrow \pi_{\theta'}(\hat{s}')$                            ▷ Select action on reconstructed $\hat{s}'$
  5: Sample $Z_{1:M} \overset{i.i.d.}{\sim} Z_\phi(s, a)$                 ▷ Samples from online critic
  6: Sample $Z_{\text{next},1:M} \overset{i.i.d.}{\sim} Z_{\phi'}(\hat{s}', a')$       ▷ Samples from target critic using reconstructed $\hat{s}'$
  7: **for** each $1 \leq i \leq M$ **do**
  8:      $Y_i \leftarrow \mathbf{f}^S_{\hat{r}, \hat{s}', \gamma}(Z_{\text{next},1:M})$             ▷ Samples from target distribution
  9: **end for**
 10: MMD$^2 \leftarrow \sum_{1 \leq i \leq M} \sum_{1 \leq j \leq M, j \neq i} [k(Z_i, Z_j) - 2k(Z_i, Y_j) + k(Y_i, Y_j)]$
 11: **return** $\nabla_\phi$MMD$^2$

---

empirical mean of the cGMMN samples. The MMD-trained model better fits the full distribution, while the regression model performs slightly better on the conditional expectation. Both methods leverage gradient information effectively, as shown by the blue curves.

**Limited data regime** However, in both supervised and reinforcement learning tasks, the assumption of unlimited data is unrealistic. Here, we explore how the performance of the two methods, cGMMN and the regression-based model, diverges when the amount of available data is restricted.

Using the same setup as before, but with a fixed number of $(x, y_{1:4})$ pairs, we assess stability by reporting the average norm of the second order derivative over the input space. For accuracy, we measure the average L2 losses between the true expected gradient and predicted gradient Results are shown in Figure 4. As can be seen, the deterministic model tends to overfit rapidly, while the cG-MMN proves more robust, maintaining better performance even with constrained data. Notably, the second-order derivative for the deterministic model escalates sharply as data becomes constrained, indicating instability in its approximation.

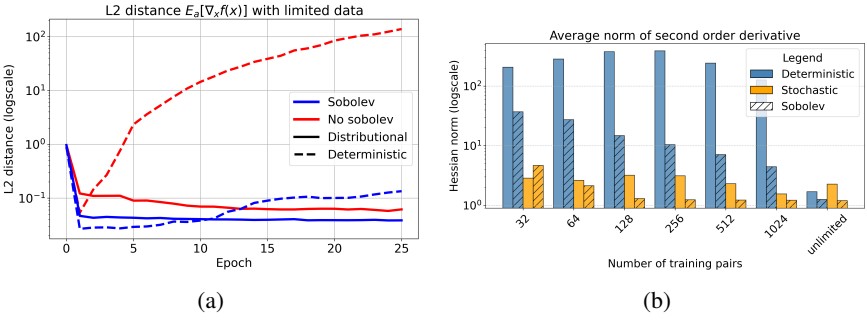

          (a)                                        (b)

Figure 4: Toy supervised learning problem. Comparison between conditonal Generative Moment Matching and deterministic regression. Left panel: training curve of L2 loss (logscale) on gradient between true conditional expectation with regression prediction and with empirical mean from cGMMN. Sobolev (blue) used gradient information to train either using MMD (full line) or L2 regression (dashed line). Right panel: average over the input space of the second order derivative (logscale) of predicted gradient from deterministic model (blue), cGMMN / stochastic (yellow) and with gradient information / Sobolev (dashed). Metrics averaged over 5 seeds.

**A common trick** Overfitting with limited data is a common issue in regression tasks. Early stopping seems an obvious solution in this case but we emphasize that it requires an evaluation criterion that is not always available (i.e in policy evaluation). Other solutions include weight regularization Krogh & Hertz (1991), dropout Srivastava et al. (2014), Bayesian neural networks Blundell et al. (2015), ensembling Chua et al. (2018), and spectral normalization Zheng et al. (2023), all of which often reduce network capacity.

To address similar issues, Fujimoto et al. (2018) proposed adding noise to the target from Eq. 3, effectively smoothing the critic. As argued by Ball & Roberts (2021), this method can be seen as indirectly acting like spectral normalization, encouraging smoother gradients and effectively reducing the magnitude of the second derivative. Appendix A.7.1 shows how noise scale impacts overfitting by inducing bias. On the other hand, we propose avoiding such assumptions by using generative modeling to add latent freedom.

## 4.2 REINFORCEMENT LEARNING

In this section, we evaluate the complete solution, including the learned world model, on several standard Mujoco environments from the BRAX library Freeman et al. (2021). We plug Algorithm. 1 into Deep Deterministic Policy Gradient, thus using an exploration policy that stores experience in a replay buffer that is then sampled uniformly from. To enhance exploration, we employ a collection of 512 actors, similar to the setup in Barth-Maron et al. (2018). The multiquadratic kernel with $h = 100$ and the biased estimator from Eq. 22 were used on every environment. Additional details regarding the architecture and hyperparameters are provided in Appendix A.8

Since most continuous control environments are deterministic in both their transition and reward functions, we applied N-step returns (with $N = 5$) to induce stochasticity as discussed in Section 2.2. The results across various environments are presented in Figure 5. We compare the two variants of DSDPG, one using action gradient and one using state-action gradient. As can be seen, both perform competitively, with at least one variant on par with the baseline DDPG across all environments. In contrast, DDPG + MMD consistently underperforms, highlighting the effectiveness of leveraging gradient information in our approach. It is worth noting, though, that the DSDPG variant using state-action gradients struggles on three out of five environments.

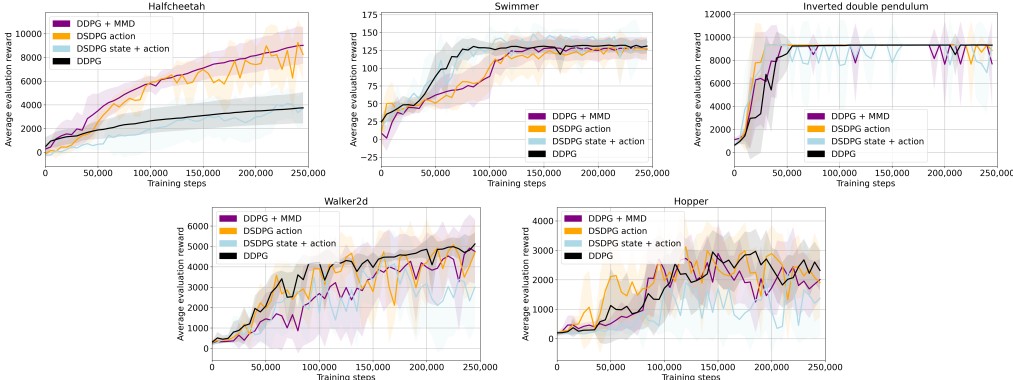

Figure 5: Comparison of Deep Deterministic Policy Gradient (DDPG) with a distributional critic trained using Maximum Mean Discrepancy on original observations (purple), our method Distribu-tional Sobolev Deterministic Policy Gradient (DSDPG) with Sobolev training on action gradients (yellow) or state-action gradients (light blue), and the standard DDPG with a deterministic critic trained on original observations (black). Rewards measured on independent evaluation with exploration noise and averaged over 5 seeds. Shaded area is +- 1 standard deviation.

For fairness, we also trained a *deterministic* critic using gradient information, following an approach similar to D'Oro & Jaskowski (2020), using an L2 loss on both the output and gradient temporal differences. However, we diverged from their method by using the same world model as for DSDPG, a cVAE, rather than their ensemble of large regressors. Results are shown in Figure 11. We were

unable to replicate their performance improvement, as the method frequently failed to converge. The cause of this discrepancy remains unclear. It could stem from the difference in world models, our modest cVAE versus their larger ensemble, or from an imbalance between the output loss and gradient loss. In MAGE (D'Oro & Jaskowski, 2020), these losses are treated separately, while in our MMD-based approach, they are handled together. Lastly, the stochastic nature of the predicted gradients might also contribute to this divergence.

When updated, the critic interacts with the environment solely through reconstructed samples from the cVAE world model. Consequently, the world model may independently influence performance, as demonstrated in Figure 12. Overall, we observe that relying on reconstructed samples tends to degrade performance, especially for the distributional critic trained with MMD but lacking gradient information. However, incorporating gradient information not only bridges this performance gap but also leads to a notable overall improvement.

## 5 RELATED WORKS

Our work extends distributional reinforcement learning (RL) (Bellemare et al., 2017) by modeling the gradients of returns, specifically in continuous action-space environments with deterministic policies. This positions our research most closely to the work of (Barth-Maron et al., 2018) which itself extended (Lillicrap et al., 2016) to the distributional setting. Since the gradient of the return is typically a multi-dimensional quantity, our approach aligns closely with studies on distributional multivariate returns (Zhang et al., 2021; Freirich et al., 2019), which emphasize the need for tractable measures of discrepancy over multi-dimensional distributions. We conceptualize our distributional critic as a generative model capable of producing actual samples of the modeled distribution, following approaches similar to (Freirich et al., 2019; Singh et al., 2022; Doan et al., 2018). This differs from methods that generate pseudo-samples (Zhang et al., 2021; Nguyen et al., 2020) or focus solely on statistics (Bellemare et al., 2017; Barth-Maron et al., 2018; Dabney et al., 2018b;a). To measure distributional discrepancy, we employ the Mean Maximum Discrepancy (MMD) (Gretton et al., 2012; Li et al., 2015; Bińkowski et al., 2021; Oskarsson, 2020), a method that has been successfully applied in distributional RL (Nguyen et al., 2020; Killingberg & Langseth, 2023; Zhang et al., 2021).

Secondly, as we explicitly model gradients using neural networks, our work can also be seen as a distributional extension of Sobolev training Czarnecki et al. (2017) which was already adapted to reinforcement learning in (D'Oro & Jaskowski, 2020). The idea of gradient modeling in value-based RL was initially explored in (Fairbank, 2008). Additionally, our approach shares connections with Physics Informed Neural Networks (PINNs) (Raissi et al., 2017), where neural networks are used to approximate physical processes and differential constraints are applied to enforce physical consistency. Uncertainty modeling in PINNs has already been considered in (Yang & Perdikaris, 2019; Daw et al., 2021).

Lastly, since the environment dynamics and reward functions are neither assumed to be differentiable nor known, we infer these quantities from true observations using a world model, similar to SVG(1) (Heess et al., 2015). To achieve this, we leverage variational inference by instantiating our world model as a conditional Variational Autoencoder (cVAE) (Kingma, 2013; Sohn et al., 2015). Although we do not use our world model to generate new data (Sutton, 1991), our approach connects to the broader literature on model-based reinforcement learning (Deisenroth & Rasmussen, 2011; Chua et al., 2018), particularly to works utilizing variational methods (Ha & Schmidhuber, 2018; Hafner et al., 2020; Zhu et al., 2024).

## 6 CONCLUSION

In this work, we introduced Distributional Sobolev Deterministic Policy Gradient (DSDPG). Our main contributions involve modeling a distribution over the output and the gradient of a random function and deriving a tractable computational scheme to do so using Maximum Mean Discrepancy (MMD). We demonstrated that the method is effective at utilizing gradient information. We empirically showed that training neural networks this way could have beneficial properties, particularly when data is scarce, while making minimal assumptions about the random function being modeled. We extended this idea to reinforcement learning by leveraging a differentiable world

model of the environment that infers gradients from observations. This approach was applied to train a distributional critic using temporal difference learning on both returns and their gradients.

This work, however, has several limitations. Future work should focus on reducing the computational cost of the current method. Currently, both policy evaluation and policy improvement require drawing several samples from the distributional critic and calculating their input gradients, leading to a significant computational burden. We suggest that more efficient inductive biases might exist, such as generating multiple samples simultaneously, offering a middle ground between deterministic noise transformation and pseudo-samples . This approach did not yield successful results in our attempts. Additionally, future research should focus on improving the design of the world model, as we found it challenging to apply the same hyperparameters across different environments. In particular, a more principled approach to weighting the KL regularization is needed.

Finally, we believe that the ideas introduced in this work could benefit other fields where uncertainty in gradient modeling is important, such as Physics-Informed Neural Networks and distillation Czarnecki et al. (2017) of generative models.

**Reproducibility Statement** Code and models will be made available upon acceptance of the paper in a public repository. This will include a README file with instructions for setting up the environment and reproducing the experiments. All datasets and environments used are publicly available.

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

# A APPENDIX

## A.1 DERIVATIVE OF A CONDITIONAL RANDOM VARIABLE WITH RESPECT TO ITS CONDITIONING VARIABLE

Here we provide an intuitive interpretation of the meaning of random variable gradients with respect to their conditioning variable. We rely on this notion in 8 and Section 3.2.

Let $y \mid x$ be a conditional random variable where $x \in \mathcal{X} \subset \mathbb{R}^n$ is the conditioning variable, and $y \in \mathcal{Y} \subset \mathbb{R}^m$. A realization of $y \mid x$ can be expressed as:

$$y(x) = g(z, x),$$

where:

- $z \in \mathcal{Z}$ is a hidden latent variable sampled from a unknown distribution $p(z)$,

- $g : \mathcal{Z} \times \mathcal{X} \to \mathcal{Y}$ is a deterministic mapping that is differentiable with respect to $x$.

The derivative of the realization $y(x)$ with respect to $x$, for a fixed latent variable $z$, is defined as:

$$\frac{\partial y}{\partial x} := \frac{\partial g(z, x)}{\partial x}.$$

Since $z \sim p(z)$, the derivative $\frac{\partial y}{\partial x}$ is itself a random variable, with its distribution induced by $p(z)$.

## A.2 PROOF OF PROPOSITION 3.1

*Let $\pi$ be an $L_\pi$-Lipschitz continuous policy, and suppose $G(s)$ and $\hat{G}(s)$ are the true and estimated distributions of the action gradients $\nabla_a Z^\pi(s, a)$ and $\nabla_a \hat{Z}(s, a)$ at $a = \pi(s)$, respectively. Then, the error between the true and estimated policy gradients is bounded by:*

$$\left\| \nabla_\theta J(\theta) - \nabla_\theta \hat{J}(\theta) \right\| \le \frac{c L_\pi}{1 - \gamma} \mathbb{E}_{s \sim d_\pi^\mu} \left[ D \left( \nabla_a Z^\pi(s, \pi(s)), \nabla_a \hat{Z}(s, \pi(s)) \right) \right],$$

*where $D$ is a discrepancy measure defined as follows:*

$$D \left( \nabla_a Z^\pi(s, \pi(s)), \nabla_a \hat{Z}(s, \pi(s)) \right) = \begin{cases} \mathcal{W}_1 \left( \nabla_a Z^\pi(s, \pi(s)), \nabla_a \hat{Z}(s, \pi(s)) \right) & \text{(Wasserstein-1 distance)}, \\ \text{MMD} \left( \nabla_a Z^\pi(s, \pi(s)), \nabla_a \hat{Z}(s, \pi(s)) \right) & \text{(Maximum Mean Discrepancy)}. \end{cases}$$

*The scaling factor $c$ is defined as:*

$$c = \begin{cases} 1 & \text{if } D \text{ is the Wasserstein-1 distance } \mathcal{W}_1, \\ \kappa^{1/2} & \text{if } D \text{ is the Maximum Mean Discrepancy (MMD).} \end{cases}$$

*Here, $\kappa = \sup_x k(x, x)$ represents the maximum value of the kernel function $k$ used in the MMD computation.*

### PROOF FOR WASSERSTEIN-1 (W1) DISTANCE

**Step 1: True and Estimated Policy Gradients**

The true policy gradient is given by:

$$\nabla_\theta J(\theta) = \frac{1}{1 - \gamma} \mathbb{E}_{s \sim d_\pi^\mu} \left[ \mathbb{E} \left[ \nabla_a Z^\pi(s, a) \big|_{a = \pi(s)} \right] \nabla_\theta \pi(s) \right],$$

The estimated policy gradient is:

$$\nabla_\theta \hat{J}(\theta) = \frac{1}{1 - \gamma} \mathbb{E}_{s \sim d_\pi^\mu} \left[ \mathbb{E} \left[ \nabla_a \hat{Z}(s, a) \big|_{a = \pi(s)} \right] \nabla_\theta \pi(s) \right].$$

**Step 3: Policy Gradient Error**

The $L^2$ norm of the difference between the true and estimated policy gradients is:

$$\left\| \nabla_\theta J(\theta) - \nabla_\theta \hat{J}(\theta) \right\| = \left\| \frac{1}{1 - \gamma} \mathbb{E}_{s \sim d_\pi^\mu} \left[ \left( \mathbb{E} \left[ \nabla_a Z^\pi(s, a) \big|_{a = \pi(s)} \right] - \mathbb{E} \left[ \nabla_a \hat{Z}(s, a) \big|_{a = \pi(s)} \right] \right) \nabla_\theta \pi(s) \right] \right\|.$$

**Step 4: Applying the Triangle Inequality and Lipschitz Continuity**

Using the triangle inequality and the Lipschitz continuity of the policy ($\|\nabla_\theta \pi(s)\| \le L_\pi$), we have:

$$\left\| \nabla_\theta J(\theta) - \nabla_\theta \hat{J}(\theta) \right\| \le \frac{1}{1 - \gamma} \mathbb{E}_{s \sim d_\pi^\mu} \left[ \left\| \mathbb{E} \left[ \nabla_a Z^\pi(s, a) \big|_{a = \pi(s)} \right] - \mathbb{E} \left[ \nabla_a \hat{Z}(s, a) \big|_{a = \pi(s)} \right] \right\| \|\nabla_\theta \pi(s)\| \right]$$

$$\le \frac{L_\pi}{1 - \gamma} \mathbb{E}_{s \sim d_\pi^\mu} \left[ \left\| \mathbb{E} \left[ \nabla_a Z^\pi(s, a) \big|_{a = \pi(s)} \right] - \mathbb{E} \left[ \nabla_a \hat{Z}(s, a) \big|_{a = \pi(s)} \right] \right\| \right].$$

**Step 5: Applying Kantorovich-Rubinstein Duality**

The Wasserstein-1 distance $\mathcal{W}_1(\mathcal{L}(X), \mathcal{L}(Y))$ between two random variables $X$ and $Y$ (with distributions $\mu$ and $\nu$, respectively) is defined as:

$$\mathcal{W}_1(\mu, \nu) = \inf_{\gamma \in \Pi(\mu, \nu)} \mathbb{E}_{(X,Y) \sim \gamma} \left[ \|X - Y\| \right],$$

where $\Pi(\mu, \nu)$ is the set of all couplings of $\mu$ and $\nu$.

By Kantorovich–Rubinstein duality Villani et al. (2009), we equivalently have:

$$\mathcal{W}_1(\mu, \nu) = \sup_{\|f\|_{\mathrm{Lip}} \leq 1} \left| \mathbb{E}_\mu[f(X)] - \mathbb{E}_\nu[f(Y)] \right|.$$

The dual formulation above holds for any $f$ with Lipschitz constant $\|f\|_{\mathrm{Lip}} \leq 1$. Choosing $f(x) = x$, we note that this function has a Lipschitz constant of 1 because $\|f(x) - f(y)\| = \|x - y\|$ satisfies the Lipschitz condition. Consequently, the difference of expectations becomes:

$$\|\mathbb{E}[X] - \mathbb{E}[Y]\| = \|\mathbb{E}_\mu[f(X)] - \mathbb{E}_\nu[f(Y)]\|.$$

Since $f$ is 1-Lipschitz, the Kantorovich–Rubinstein duality ensures that this is bounded by the Wasserstein-1 distance:

$$\|\mathbb{E}[X] - \mathbb{E}[Y]\| \leq \mathcal{W}_1(\mathcal{L}(X), \mathcal{L}(Y)).$$

Let $X = \nabla_a Z^\pi(s, \pi(s))$ and $Y = \nabla_a \hat{Z}(s, \pi(s))$. The difference of their expectations is:

$$\left\| \mathbb{E}\left[\nabla_a Z^\pi(s, \pi(s))\right] - \mathbb{E}\left[\nabla_a \hat{Z}(s, \pi(s))\right] \right\|.$$

Using the argument above, this difference is bounded by the Wasserstein-1 distance between the distributions of $X$ and $Y$:

$$\left\| \mathbb{E}\left[\nabla_a Z^\pi(s, \pi(s))\right] - \mathbb{E}\left[\nabla_a \hat{Z}(s, \pi(s))\right] \right\| \leq \mathcal{W}_1\left(\nabla_a Z^\pi(s, \pi(s)), \nabla_a \hat{Z}(s, \pi(s))\right).$$

**Step 6: Conclusion**

Combining the results from the previous steps, we established that the $L^2$ norm of the difference between the true and estimated policy gradients can be bounded as follows:

$$\left\| \nabla_\theta J(\theta) - \nabla_\theta \hat{J}(\theta) \right\| \leq \frac{L_\pi}{1 - \gamma} \mathbb{E}_{s \sim d_\pi^\mu} \left[ \mathcal{W}_1\left(\nabla_a Z^\pi(s, \pi(s)), \nabla_a \hat{Z}(s, \pi(s))\right) \right].$$

PROOF FOR MAXIMUM MEAN DISCREPANCY (MMD)

The first steps are identical to the proof for the Wasserstein-1 distance.

**Step 1: Bounding the Expectation Difference Using MMD**

The MMD between the distributions of $\nabla_a Z^\pi(s, \pi(s))$ and $\nabla_a \hat{Z}(s, \pi(s))$ is defined as:

$$\mathrm{MMD}^2(\nabla_a Z^\pi(s, \pi(s)), \nabla_a \hat{Z}(s, \pi(s))) = \mathbb{E}_{X, X' \sim \mathcal{L}(Z)}[k(X, X')]$$
$$+ \mathbb{E}_{Y, Y' \sim \mathcal{L}(\hat{Z})}[k(Y, Y')]$$
$$- 2\mathbb{E}_{X \sim \mathcal{L}(Z), Y \sim \mathcal{L}(\hat{Z})}[k(X, Y)],$$

where $k$ is the kernel function, and $\mathcal{L}(Z)$ denotes the distribution of $\nabla_a Z^\pi(s, \pi(s))$, while $\mathcal{L}(\hat{Z})$ denotes the distribution of $\nabla_a \hat{Z}(s, \pi(s))$.

**Step 2: Relating Policy Gradient Error to MMD** For a linear kernel $k(x, y) = x^\top y$, the MMD simplifies to:

$$\mathrm{MMD}_{\mathrm{linear}}(\nabla_a Z^\pi(s, \pi(s)), \nabla_a \hat{Z}(s, \pi(s))) = \left\| \mathbb{E}\left[\nabla_a Z^\pi(s, \pi(s))\right] - \mathbb{E}\left[\nabla_a \hat{Z}(s, \pi(s))\right] \right\|.$$

Substituting this result back, we have:

$$\left\| \nabla_\theta J(\theta) - \nabla_\theta \hat{J}(\theta) \right\| \leq \frac{L_\pi}{1 - \gamma} \mathbb{E}_{s \sim d_\pi^\mu} \left[ \mathrm{MMD}_{\mathrm{linear}}(\nabla_a Z^\pi(s, \pi(s)), \nabla_a \hat{Z}(s, \pi(s))) \right].$$

**Step 3: Extending to Non-Linear Kernels**

For non-linear kernels, the MMD measures differences in higher-order statistics. Using the Cauchy-Schwarz inequality in the RKHS, we have:

$$\left\| \mathbb{E}\left[ \nabla_a Z^\pi(s, \pi(s)) \right] - \mathbb{E}\left[ \nabla_a \hat{Z}(s, \pi(s)) \right] \right\| \leq \kappa^{1/2} \text{MMD}(\nabla_a Z^\pi(s, \pi(s)), \nabla_a \hat{Z}(s, \pi(s))),$$

where $\kappa = \sup_x k(x, x)$ is the maximum value of the kernel function.

Substituting this bound back:

$$\left\| \nabla_\theta J(\theta) - \nabla_\theta \hat{J}(\theta) \right\| \leq \frac{L_\pi \kappa^{1/2}}{1 - \gamma} \mathbb{E}_{s \sim d_\pi^\mu} \left[ \text{MMD}(\nabla_a Z^\pi(s, \pi(s)), \nabla_a \hat{Z}(s, \pi(s))) \right].$$

**Step 4: Conclusion**

Combining the results from the previous steps, we established that the $L^2$ norm of the difference between the true and estimated policy gradients can be bounded as follows:

$$\left\| \nabla_\theta J(\theta) - \nabla_\theta \hat{J}(\theta) \right\| \leq \frac{L_\pi \kappa^{1/2}}{1 - \gamma} \mathbb{E}_{s \sim d_\pi^\mu} \left[ \text{MMD}(\nabla_a Z^\pi(s, \pi(s)), \nabla_a \hat{Z}(s, \pi(s))) \right].$$

## A.3 DISTRIBUTIONAL SOBOLEV OPERATOR

In Section 3.2, we defined the new operator:

$$\mathbf{f}_{s,a,r,s',\gamma}^{S_a}(x) = \left[ \begin{array}{c} f_{s,a,r,s',\gamma}^{\text{return}}(x) \\ f_{s,a,r,s',\gamma}^{\text{action}}(x) \end{array} \right], \tag{25}$$

where

$$f_{s,a,r,s',\gamma}^{\text{return}}(x) = r + \gamma x^{\text{return}}, \tag{26}$$

and

$$f_{s,a,r,s',\gamma}^{\text{action}}(x) = \frac{\partial}{\partial a} r(s, a) + \gamma \frac{\partial s'}{\partial a} \left( \frac{\partial}{\partial s'} x^{\text{return}} + \frac{\partial a'}{\partial s'} x^{\text{action}} \right). \tag{27}$$

We now demonstrate how to obtain the result from Eq. 27. We first assume that samples from the transition-reward distributions are differentiable with respect to state and action. We start from Eq. 26 and write down the action-gradient. We abuse notation and consider $r, s$ and $x^{\text{return}}$ as explicit samples conditioned on $(s, a)$

$$
\begin{aligned}
\frac{\partial}{\partial a}\left(r + \gamma x^{\text{return}}\right) &= \frac{\partial}{\partial a}\left(r(s, a) + \gamma z(s, a)\right) \\
&= \frac{\partial}{\partial a} r(s, a) + \frac{\partial s'}{\partial a} \left( \frac{\partial}{\partial s} z(s, \pi(s')) \Big|_{s=s'} + \frac{\partial \pi(s')}{\partial s'} \frac{\partial}{\partial a} z(s', a) \Big|_{a=\pi(s')} \right)
\end{aligned} \tag{28}
$$

We identify that $\frac{\partial}{\partial s} z(s, \pi(s')) \Big|_{s=s'}$ is equivalent to $\frac{\partial}{\partial s'} x^{\text{return}}$ and $\frac{\partial}{\partial a} z(s', a) \Big|_{a=\pi(s')}$ is equivalent, by definition, to $x^{\text{action}}$

## A.4 CONDITIONAL VARIATIONAL AUTO-ENCODERS

A principled invertible generative model can be obtained from a Variational Auto-Encoder (VAE) (Kingma, 2013). More interestingly for us are conditional VAE Sohn et al. (2015) which we briefly introduce.

A Conditional Variational Autoencoder (cVAE) is a generative model that learns to generate new samples from a distribution conditioned on given input information. In our case, the cVAE models the distribution of next states and rewards conditioned on current states and actions.

Formally, the cVAE consists of two components:

- **Encoder**: The encoder $q_\zeta(\varepsilon \mid s', r; s, a)$ maps the observed next state $s'$ and reward $r$, conditioned on the current state-action pair $(s, a)$, to a latent variable $\varepsilon$, typically modeled as a Gaussian distribution with diagonal covariance matrix:

$$q_\zeta(\varepsilon \mid s', r; s, a) = \mathcal{N}(\varepsilon; \mu_\zeta(s', r, s, a), \sigma_\zeta^2(s', r, s, a) \odot I). \tag{29}$$

- **Decoder**: The decoder $p_\psi(s', r \mid \varepsilon; s, a)$ reconstructs the next state $s'$ and reward $r$ from the latent variable $\varepsilon$, conditioned on the current state-action pair $(s, a)$.

The objective of a cVAE is to maximize the Evidence Lower Bound (ELBO), which balances accurate reconstruction of the input with a regularization term that ensures the learned posterior distribution remains close to the prior distribution. In this work we adopted $\beta - \text{VAE}$ Higgins et al. (2017) which put a weight $\neq 1$ on the KL regularization term. The objective is as follows

$$\begin{aligned}
\mathcal{L}_{\text{cVAE}}(\zeta, \psi) &= \mathbb{E}_{q_\zeta(\varepsilon \mid s', r; s, a)} \left[ \log p_\psi(s', r \mid \varepsilon; s, a) \right] \\
&\quad - \lambda_{\text{KL}} \times D_{\text{KL}} \left( q_\zeta(\varepsilon \mid s', r; s, a) \parallel \mathcal{N}(0, I) \right).
\end{aligned} \tag{30}$$

The first term encourages faithful reconstruction of the next state and reward, while the second term regularizes the posterior distribution to remain close to a standard Gaussian prior. Assuming the decoder $p_\psi(s', r \mid \varepsilon; s, a)$ is Gaussian with a fixed variance, the reconstruction term reduces to an L2 loss, which can be estimated using the difference between the reconstructed samples and the true samples.

Assuming the encoder parametrizes a Gaussian with diagonal covariance and that the prior is also Gaussian with identity covariance and zero mean, the KL divergence can be estimated from encoded input samples as

$$D_{\text{KL}}(\zeta) = \mathbb{E}_{(s, a)} \left[ \frac{1}{2} \sum_{j=1}^{d} \left( 1 + \log(\sigma_{\zeta, j}^2(s, a)) - \mu_{\zeta, j}^2(s, a) - \sigma_{\zeta, j}^2(s, a) \right) \right]. \tag{31}$$

### A.5 Sampling from auto-encoding model

Following the approach from Heess et al. (2015), we propose to use a world model that can essentially work in two ways: imagination or inference from real observations. Imagination involves using samples from the generative model to use in Eq. 24 whereas inference uses actual observations to infer the latent variable $\varepsilon$. The inference approach uses the encoder and generator to reconstruct samples. In this section we aim to show that both are valid ways to get samples to differentiate.

We seek to estimate samples from the function on random variables we write as $g(s, a, s', r, \varepsilon_w)$. This implies sampling from $p(s', r, \varepsilon_w \mid s, a)$. Imagination allows a forward process $\varepsilon_w \mid a, s \to \hat{s}', \hat{r}$. But we seek to evaluate $g$ from actual observations $s', r \mid s, a \to \varepsilon_w$. This is in essence the same idea as developed in Heess et al. (2015) except we work on tuple of next state and reward instead of state and action. Moreover, our policy is assumed to be deterministic.

We want to show that imagination and inference are equivalent such that

$$p(s', r, \varepsilon_w \mid s, a) = \underbrace{p(\varepsilon_w) p(s', r \mid s, a, \varepsilon_w)}_{\text{imagination/forward}} = \underbrace{p(s', r \mid s, a) p(\varepsilon_w \mid s, a, s', r)}_{\text{inference/reconstruction}} \tag{32}$$

Using Bayes' theorem, we have:

$$p(\varepsilon_w \mid s, a, s', r) = \frac{p(s', r \mid s, a, \varepsilon_w) p(\varepsilon_w \mid s, a)}{p(s', r \mid s, a)} \tag{33}$$

Assuming $\varepsilon_w$ is independent of $s$ and $a$, we simplify $p(\varepsilon_w \mid s, a)$ to $p(\varepsilon)$, thus:

$$p(\varepsilon_w \mid s, a, s', r) = \frac{p(s', r \mid s, a, \varepsilon_w) p(\varepsilon_w)}{p(s', r \mid s, a)} \tag{34}$$

Rearranging to isolate $p(s', r \mid s, a)$, we get:

$$p(s', r \mid s, a) = \frac{p(s', r \mid s, a, \varepsilon_w) p(\varepsilon_w)}{p(\varepsilon_w \mid s, a, s', r)} \tag{35}$$

Substitute this expression back into $p(s', r \mid s, a) p(\varepsilon_w \mid s, a, s', r)$:

$$p(s', r \mid s, a) p(\varepsilon_w \mid s, a, s', r) = p(\varepsilon_w) p(s', r \mid s, a, \varepsilon_w) \tag{36}$$

$$\blacksquare$$

## A.6   INFERRING GRADIENTS FROM RECONSTRUCTED SAMPLES

Here we demonstrate visually how, using a cVAE trained on the toy supervised task introduced in Section 4.1, we can infer gradients from reconstructed samples. This exemplifies the idea of a cVAE world model applied in Section 4.2. Result is show in Figure 6. As can be seen on the left panel, the reconstruction is near perfect while the gradient inferred from those reconstructed samples matches the true gradient to some extent. Indeed, we see that without using gradient information for training the cVAE cannot disentangle ambiguous locations such as $x = 0$ but especially $x = \pi$ where the gradient collapses to the conditional expectation at that point.

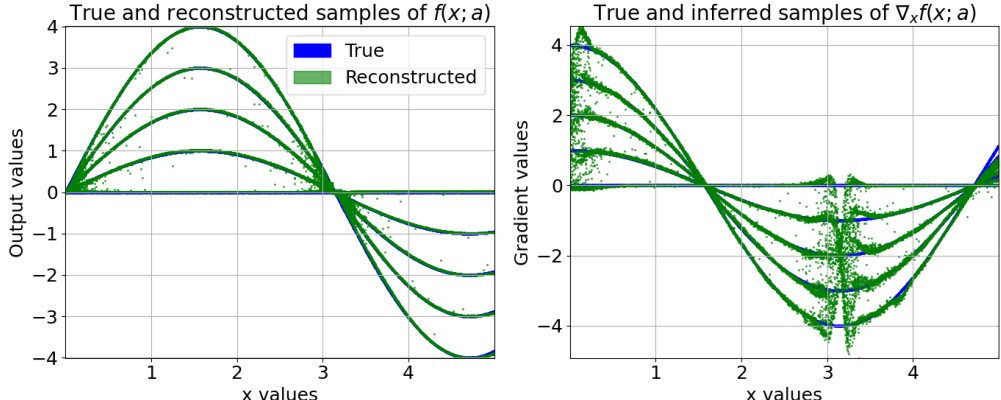

Figure 6: Gradient inference from reconstructed samples in the same toy problem as described in Section 4.1. The left panel compares the true distribution (blue) with reconstructed samples (green) after passing them through the encoder and decoder. The right panel shows a comparison of true gradients with inferred gradients on the reconstructed samples, where the gradient from the latent variable is blocked.

## A.7   TOY SUPERVISED

The Conditional Generative Moment Matching and regression used the same architecture except for some noise of dimension 10 drawn from $\mathcal{N}(0; I)$ concatenated to the input for the cGMMN. For each pair $(x, y_{1:4})$, four samples were drawn from the generator. Both were trained using Rectified Adam Liu et al. (2019) optimizer with a learning rate of $1 \times 10^{-3}$ and $(\beta_0, \beta_1) = (0.5, 0.9)$. Neural network is a simple MLP with 2 hidden layers of 256 neurons and Swish non-linearities (Ramachandran et al., 2017).

Maximum Mean Discrepancy (MMD) was estimated using a mixture of RBF kernels with bandwidths $\sigma_i$ from the set $\{\sigma_1, \sigma_2, \ldots, \sigma_7\} = \{0.01, 0.05, 0.1, 0.5, 1, 10, 100\}$. We used the biased estimator from Eq. 22.

The equation for a mixture of RBF kernels is given by:

$$k^{\text{mix}}(x, y) = \sum_{i=1}^{7} \exp\left(-\frac{\|x - y\|^2}{2\sigma_i^2}\right).$$

(37)

The evaluation kernel we used was the Rational Quadratic $k_\alpha^{\text{RQ}}$ with $\alpha = 1$ with

$$k_\alpha^{\text{RQ}}(x, y) = \left(1 + \frac{\|x - y\|^2}{2\alpha}\right)^{-\alpha}$$

(38)

Regarding the dataset, the $(x, y_{1:4})$ pairs were drawn with $x \sim \mathcal{U}[0; 5]$ and $a$ was draw from $\{0, 1, 2, 3, 4\}$ with replacement. In the limited data regime, the pairs $(x, y_{1:4})$ were drawn once and stayed fix. The batch size was thus equal to the number of points in the dataset. In the unlimited

data regime 256 new pairs were drawn for each batch. Every experiment was ran for 25 000 batch sampled and thus the same number of SGD steps.

### A.7.1 ADDING NOISE

Inspired by Fujimoto et al. (2018), we added some independent noise on $x$ for each $(x, y_{1:4})$. Noise scale $\sigma$ was in $\{0.01, 0.1, 0.5\}$. For each new batch sampled it was sampled from a standard Gaussian $\eta \sim \mathcal{N}(0; \sigma^2)$ and added as $\tilde{x} = x + \eta$

In Figure 7-8, we can see the impact of the various noise scales on the predictions of the deterministic regression. As can be seen, adding noise on $x$ as a positive effect in terms of stabilizing the gradient but it induces an bias that grows with the scale of the noise. Moreover, as discussed in Section 4.1, this noise depend on the application and makes strong assumption about the function to learn. The stabilizing effect of additive noise can further be seen in Figure 9-10 where both the L2 loss and average second order derivative are displayed as function on the number of sampled locations.

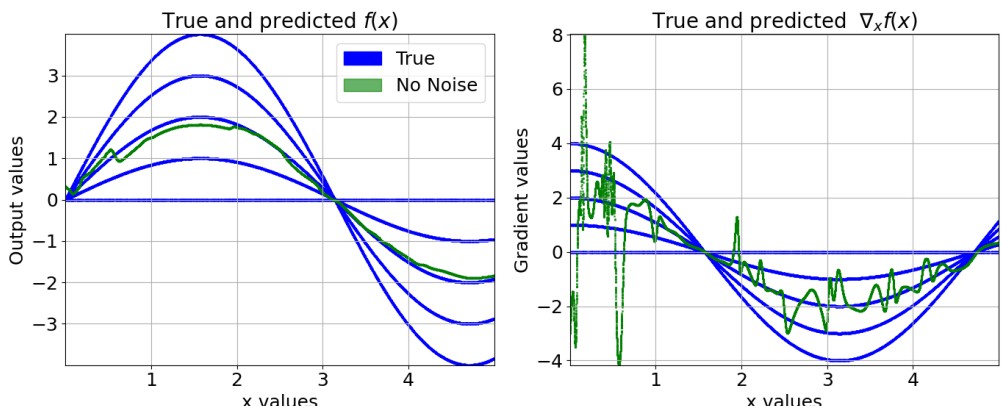

Figure 7: Toy supervised learning problem. Comparison of samples from the true five-mode distribution with predictions made by a deterministic model trained with L2 loss (green). The output space is shown on the left, and the gradient space on the right. Results obtained after 25,000 training steps.

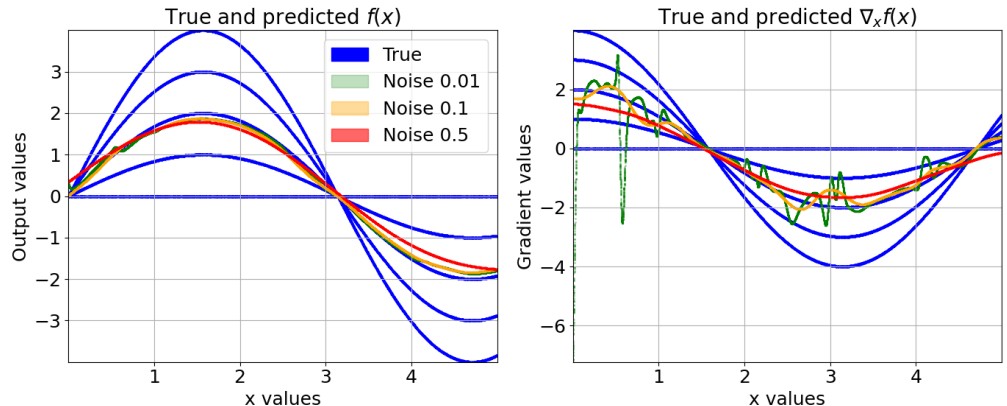

Figure 8: Toy supervised learning problem. Comparison between true samples from the distribution of five modes and deterministic models trained with varying levels of additive noise on their input data. Low level of noise (green), medium level of noise (orange), high level of noise (red). Results obtained after 25,000 training steps.

### A.8 REINFORCEMENT LEARNING EXPERIMENTS

The experiments were ran in BRAX Freeman et al. (2021), a JAX Bradbury et al. (2018) re-implementation of common Mujoco environments (Todorov et al., 2012). We took advantage of

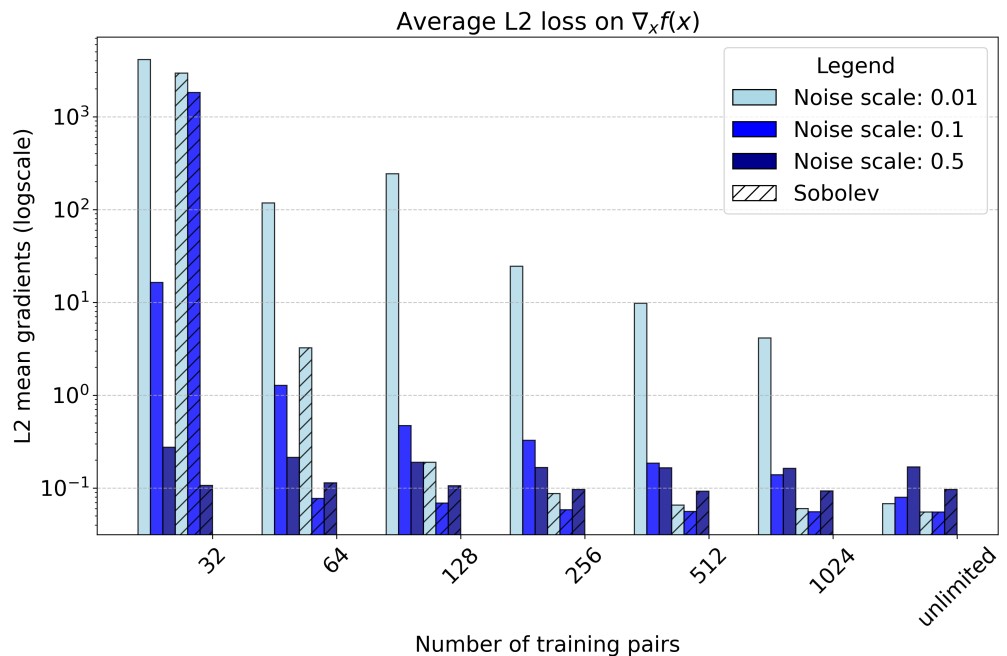

Figure 9: Toy supervised learning problem. Comparison of the L2 loss between the predicted gradient and the conditional expectation of the true distribution. Different scales of additive noise on the input are compared: low noise (light blue), medium noise (medium blue), and high noise (dark blue), alongside Sobolev training (dashed). Results are shown after 25,000 training steps.

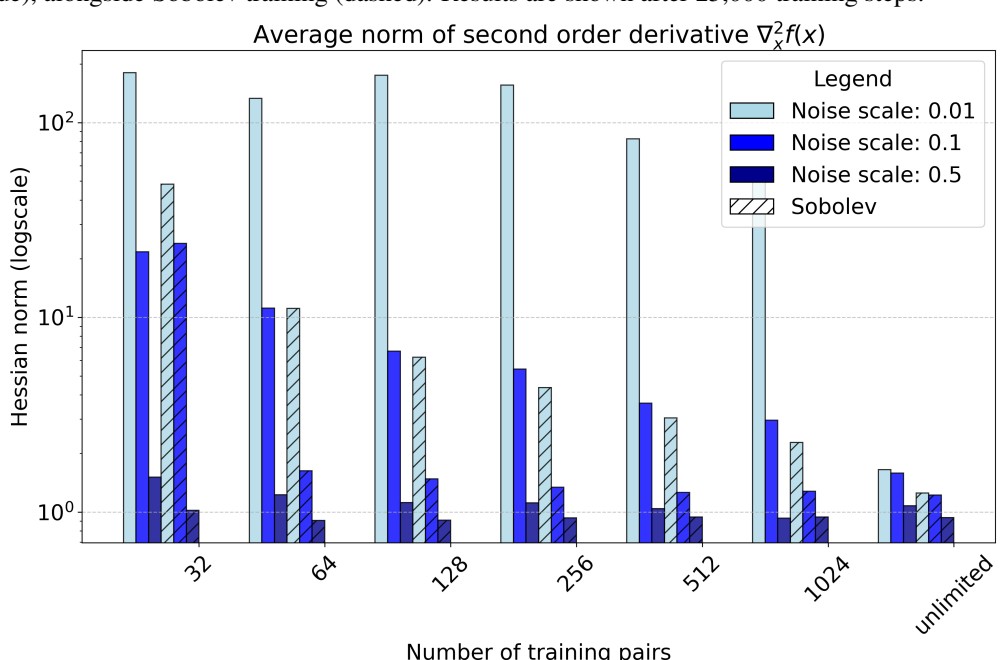

Figure 10: Toy supervised learning problem. Comparison of the average second order derivative norm over the input space. Different scales of additive noise on the input are compared: low noise (light blue), medium noise (medium blue), and high noise (dark blue), alongside Sobolev training (dashed).

BRAX's high parallelizability to have 512 actors running the exploration in parallel. Their experiences were pushed in a uniform replay buffer. The procedure described in Algorithm 1 was plugged into DDPG (Lillicrap et al., 2016).

Now we describe the architectures, optimizers and other hyperparameters of the full Distributional Sobolev Deterministic Policy Gradient algorithm.

Table 1: Hyperparameters for the DDPG and DSDPG experiments on Brax environments

| Item | Value |
|------|-------|
| Discount $\gamma$ | 0.99 |
| Polyak averaging $\tau$ | 0.005 |
| Buffer size | $10^6$ |
| Exploration noise scale | 0.1 |
| Critic learning rate | $3 \times 10^{-4}$ |
| Policy learning rate | $3 \times 10^{-4}$ |
| cVAE learning rate | $6 \times 10^{-4}$ |
| cVAE KL weight | 0.1 |
| cVAE latent dim | $|\mathcal{S}| + 1$ |
| Critic input noise dim | 50 |
| Number of samples MMD | 50 |

**Policy network**    Policy network is a MLP with 2 hidden layers of 256 neurons. In order to improve gradient flow, skip connections from the input $s$ to hidden layers' input were used as well as residual connection. The non-linearity was Swish (Ramachandran et al., 2017). Final activations are mapped to the output space using a linear transformation followed by a $\tanh$ non-linearity. The policy network was optimized using the Rectified Adam Liu et al. (2019) with a learning rate of $3 \times 10^{-4}$.

**Crtic network**    Critic network architecture is almost the same as for the policy network. Importantly, it is kept constant for experiments using normal DDPG and DSDPG apart from noise concatenated on the input $[s; a]$. The network is a MLP with 256 neurons, skip connections from the input, residual connection and Swish activations. No non-linearity was applied on the output after the last linear layer. The critic network is optimized using Rectified Adam Liu et al. (2019) with $(\beta_1, \beta_2) = (0.9, 0.999)$ and a learning rate of $3 \times 10^{-4}$.

**Conditional VAE world model**    Both the encoder and decoder are MLPs with 3 hidden layers, each containing 512 neurons. Skip connections are applied from the input to each hidden layer, and Layer Normalization (Ba, 2016) is used after the non-linearity activations to normalize the hidden layers. The cVAE was optimized using Adam, as we observed using RAdam to systematically diverge, $(\beta_1, \beta_2) = (0.9, 0.999)$ and a learning rate of $3 \times 10^{-4}$. As explained in A.4, we used a $\beta - \text{VAE}$ where the weight on KL divergence was set to 0.1 as we found it to work well on most environments.

The prior is a fixed standard Gaussian $\mathcal{N}(0, I)$ with a latent dimension equal to the size of the random variable being modeled, which is $|\mathcal{S}| + 1$ for $(s', r)$. Following D'Oro & Jaskowski (2020); Zhu et al. (2024), the cVAE predicts the difference between the current and next observation, $\delta_s = s' - s$ which is then added back to $s$, along with the reward $r$.

**Conditional Generative Moment Matching**    For the distributional critic, noise vectors were concatenated with the state-action pairs $(s, a)$ and passed through the same architecture as the deterministic critic. The noise dimension was set to 50. For each state-action pair, 50 samples were drawn to update both the critic and the policy. The multiquadratic kernel $k_h^{\text{MQ}}(x, y) = -\sqrt{1 + h^2 \|x - y\|_2^2}$ Killingberg & Langseth (2023) was used, with the kernel parameter $h$ set to 100.

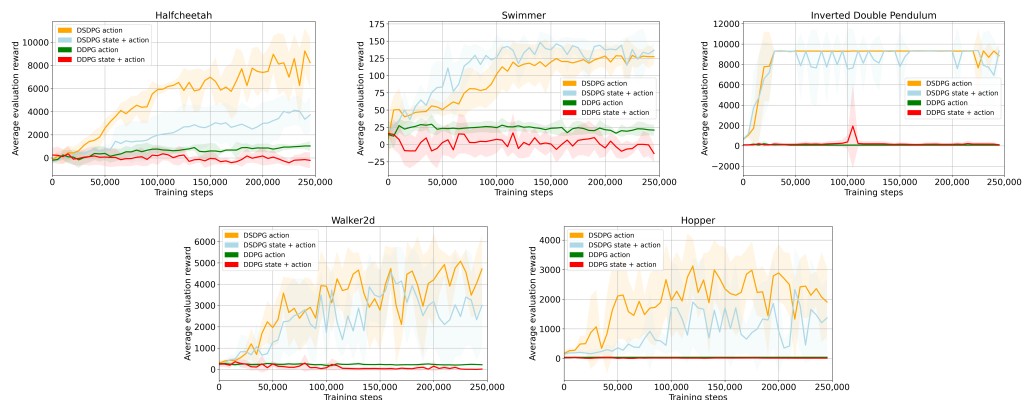

Figure 11: Comparison of our method DSDPG with action gradient training (yellow) or state-action gradient training (light blue) with deterministic Sobolev training using L2 loss.

### A.8.1 COMPARISON DETERMINISTIC AND DISTRIBUTIONAL SOBOLEV TRAINING

### A.8.2 INFLUENCE OF WORLD MODEL ON PERFORMANCE

When updated, the critic interacts with the environment exclusively through reconstructed samples from the cVAE world model. Consequently, the quality of the world model may independently influence performance, as shown in Figure 12. Overall, we observe that using reconstructed samples from the world model generally has a negative impact on performance, particularly for the distributional critic trained with MMD but without gradient information. However, gradient information not only compensates for this gap but also leads to an overall improvement in performance.

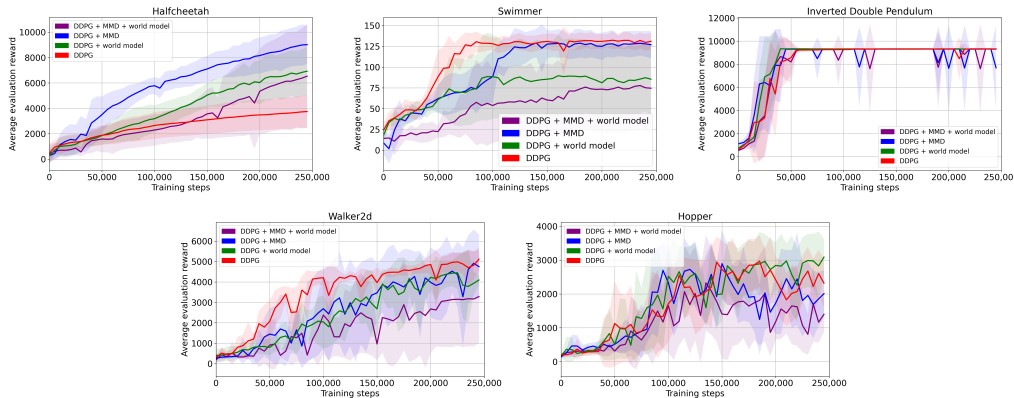

Figure 12: Impact of the world model on the performance of DDPG with a deterministic critic and a distributional critic trained using MMD. The results compare DDPG (red), DDPG trained on reconstructed samples from the cVAE (green), DDPG with a distributional critic (blue), and DDPG with a distributional critic trained on reconstructed samples (purple).

