# OpenReview forum: "Distributional Sobolev reinforcement learning"
_ICLR.cc/2025/Conference — Submitted to ICLR 2025_

### Official Review · Reviewer_HhAu · 2024-10-26

**Soundness:** 3
**Presentation:** 2
**Contribution:** 2
**Rating:** 5
**Confidence:** 4

**Summary:**

This paper introduces a novel approach to distributional reinforcement learning, Distributional Sobolev RL, which models both the distribution of random returns and the action-gradients. The paper also empirically demonstrate the effectiveness of proposed algorithm. While incorporating Sobolev training is an interesting and promising idea, I have some concerns regarding the current manuscript.

**Strengths:**

The motivation is clear, and the idea of learning distribution over action gradients is good.

**Weaknesses:**

- **Experiments**: There are several issues that need to be clarified in the toy experiments. For example, the red and blue dashed lines in Figures 3 and 4 are confusing. In Figure 1, it is difficult to distinguish the blue and green lines. For MuJoCo experiments, I have big concerns about the **fairness comparison**. The authors only present the performance results after $0.25\times10^6$ training steps, while the default setting is  $1\times10^6$. The potential cause I guess is that DSDPG is computationally cost, but this adjustment seems to benefit DSDPG since it is a model-based algorithm utilizing pre-trained transition information, which helps faster convergence. In contrast, DDPG is a model-free algorithm that is not sample-efficient, particularly in the early stages.  Furthermore, using actor-critic-based methods to demonstrate the advantage of incorporating gradient information into distributional modeling may not be ideal since the decision-making is based on the actor, while the critic (return distribution) is primarily used to train the actor network.

- **Random state-action Sobolev return**: First, It is unclear whether Sobolev returns are actually used for decision-making. Second, Equation (15) is confusing, and it appears to be missing a term involving $\frac{\partial s'}{\partial s}\frac{\partial s}{\partial a} x$.  Third, as the Sobolev return is defined as the derivative of the return with respect to the state, I have a concern regarding the handling of complex state spaces, such as pixel-based states in Atari games. The computational burden will significantly increase as the dimension of the state grows. Since the current experiments are implemented on Mujoco, where the state and action are vectors, I am curious about how well DSRL will perform in more complex environments.



- **Discussion with other methods:**  The discussion of related methods is limited. It would be beneficial to expand on this, especially regarding the connections to existing distributional RL approaches (QR-DQN, C51). The use of MMD minimization for learning the distributional Bellman operator is inspired by MMD-DQN [1], but this connection should be discussed in more detail. Additionally, tuning the bandwidth parameter $h$ in the multiquadratic kernel is crucial since kernel-based methods are sensitive to bandwidth selection.

[1] Distributional Reinforcement Learning via Moment Matching. AAAI 2021.

**Questions:**

Q1: I have concerns regarding sample-based distributional RL. While some works utilize samples to represent return distributions, these methods are not mainstream. One significant drawback is the computational inefficiency associated with using generative models to represent return distributions. Furthermore, sample-based methods involving maximum likelihood estimation and specific distributional parameterization may suffer from model misspecification, making it difficult to capture the distributional Bellman equation.

Q2: Have the authors considered using diffusion models to model transition probabilities, given their impressive empirical performance?

Q3: Have the authors considered implementing Distributional Sobolev RL on top of value-based RL?

---

> ### Author Response · Authors · 2024-11-23
>
> Thank you for taking the time to thoroughly review our work and provide valuable insights and constructive feedback.
>
> Here we try to address your questions and concerns.
>
> # Experiments
> * **Figures clarification**
>    - Figures 3 and 4: we clarified (in red) the meaning of each curve. Sobolev (blue) indicates we use gradient information to learn the function or distribution, No Sobolev (red) only uses the scalar information from the dataset. More specifically about the dashed lines: it is a setup where a similar architecture to the *distributional* setting is used except it is simply L2 regression.
>    - Figure 1: the blue dots are indeed hidden behind the other curves. Blue are samples from the true 5 modes of the distribution over functions. You can find clearer examples in Figure 7 and 8 (in the Appendix). We will make sure to change the Figure to something clearer if needed.
> * **Fairness**  we acknowledge we only showcase our method on $0.25 \times 10^6$ training steps. However, as we wrote in the experiment section, we use 512 actors to speed up convergence similarly to [1]. Furthermore, as you can see most methods reached convergence before hitting the maximum number of steps.
> * **Pre-trained transition information** we do not pre-train our world model but learn it concurrently to the policy and critic.
> * **Incorporating gradient information into distributional modelling** we argue this is the idea scenario as policy improvement is carried out from the critic's action gradient. Learning a more accurate gradient as in [2] and then a distribution over that gradient is done to help policy improvement by more accurate policy evaluation.
>
> # Random state-action Sobolev return
>
> * ***It is unclear whether Sobolev returns are actually used for decision making*** They are modeled during policy evaluation. The modeled Sobolev return distribution is then sampled from during the policy improvement step, where the action-gradient part is used to provide an improvement direction to the policy. We further clarify our method is value-based. This process is described in equation 8 except in practice we draw samples from a replay buffer.
> * **Eq. 15** We clarified equation 15 and put a small derivation in the appendix.
> * ***Handling of complex state-spaces*** indeed learning world-model for complex state spaces might not be trivial and the estimated state-gradient might not be accurate. It is a limitation that we do not tackle in this work but we point to methods that use variational inference and cVAEs to do transition modelling in a latent space instead of the state-space directly [4]. This could be the topic of further work.
>
> # Discussion with other methods
> * The works you mentioned are cited in the Section 3.3 after the *Distributional Sobolev critic* paragraph. As we wrote, neither QR-DQN nor C51 were applicable to learning a distribution over gradient. Our framework is similar to learning a distribution over multi-variate returns [5]. Learning quantiles or bins isn't tractable over many dimensions thus we favored a method based on generative modelling and design our distributional Sobolev critic as a geneative model that minimizes a distributional discrepancy (MMD) that can be tractacbly estimated via samples.
> * **Kernel tuning**: Indeed, tuning the kernel and its hyperparameters is critical. We used a value we found work well on most environments which is also the same as in [3]
>
> # Questions
> * **Computational cost** Indeed, modeling the return distribution using generative modeling is more computationally expensive than via pseudo-samples or quantiles for example. This is a property we acknowledge in the conclusion. One path towards reducing that issue would be to share the forward pass between the K samples. We investigated that idea but did not succeed. Furthermore, as far as we are aware, there is no other obvious way to model a random gradient (using Sobolev inductive bias we mentionned in the submission) than via generative modelling and the reparametrization trick.
> * **Diffusion** Diffusion is a possible avenue for the world model. However, the computational cost will grow dramatically. As we need to differentiate the compute graph of the generative model with respect to the conditioning variable. Diffusion models implicitly build very deep compute graphs. Furthemore, our method for gradient estimation (of the environment) relies on inference. Switching to diffusion would probably mean we would rather use the world-model to imagine new samples (that would be differentiable) instead of reconstructing/inferring.
> * ***Have the authors considered implementing Distributional Sobolev RL on top of value-based RL?*** We are confused as we did implement Distributional Sobolev training on a value-based baseline (DDPG).
>
> We hope the revisions address your concerns and provide further clarity on the novelty and contributions of our method. Thank you once again for your time and constructive comments.

---

> > ### Author Response · Authors · 2024-11-23
> > **references**
> >
> > [1] Distributed Distributional Deterministic Policy Gradients https://arxiv.org/abs/1804.08617
> >
> > [2] How to Learn a Useful Critic? Model-based Action-Gradient-Estimator Policy Optimization
> >
> > [3] The Multiquadric Kernel for Moment-Matching Distributional Reinforcement Learning
> >
> > [4] MASTERING ATARI WITH DISCRETE WORLD MODELS
> >
> > [5] M3DQN

---

### Official Review · Reviewer_o3YT · 2024-10-26

**Soundness:** 1
**Presentation:** 1
**Contribution:** 1
**Rating:** 1
**Confidence:** 4

**Summary:**

This paper proposed a so-called Distributional Sobelev Deterministic Policy gradient, where the proposed method uses a distributional critic loss while incorporating the gradient of a random variable. Particularly, in RL scenarios, the authors also leverage a differential world model of the environment in order to infer gradients from observations. They also conducted some experiments on the toy supervised learning tasks and several Mujoco envs.

**Strengths:**

The proposed algorithm seems to incorporate different kinds of methods, such as Sobolev learning and conditional VAE.

**Weaknesses:**

**A poor motivation**. In the introduction part, it seems that the authors try to motivate the paper to improve the (stability of) policy gradient algorithm by incorporating the gradient information in the policy optimization. This is a very straightforward and even trivial idea. Also, the authors tried to develop a distributional version of algorithms that is orthogonal to the existing algorithm and has already been studied before, such as [1]. It was very confusing to understand the motivation behind it, but I ended up being frustrated.

**No methodological contribution**. Distributional RL and Sobolev are already known methods, and I do not think it is meaningful to do such kind of combination research. Also, the proposed method is constrained. For example, the paper directly limits the scope to a deterministic policy gradient method starting from Eq.(2). The gradient to be incorporated into the training requires the differentiability of the environment. To this end, the authors proposed a world model by CVAE, but it is not practical for real and potentially complicated environments.

**Poor Writing**. It is very hard to understand what the authors want to express. Many expressions are very inaccurate, and notation usage is sloppy. For example, what is the likelihood estimation issue in Line 215? How to define the derivative of a random variable over a in Eq.8? Note that the return Z is a random variable conditionally over the policy on s and a. I am confused to understand this point. The reference of Czarnecki et al., 2017 is also not complete, making me feel the writing of this paper is very non-professional. Z^S in Eq. 11 is not consistent with Z_S, which occurs many times.

**The experiments are trivial and limited**. It is not clear if it is necessary to have experiments on supervised learning tasks since the focus of this paper is RL. The empirical improvement in Mujoco is insignificant, making me feel that the proposed method is even not sound.


[1] Distributed Distributional Deterministic Policy Gradients (ICLR 2018)

**Questions:**

Please see Weakness.

---

> ### Author Response · Authors · 2024-11-23
>
> Thank you for taking the time to thoroughly review our work and provide valuable insights and constructive feedback.
>
> Here we try to address your questions and concerns.
>
> * **Motivation** we clarified our theoretical motivation by adding Proposition 3.1 in the approach section as well as some more explanations. We do not use the gradient information in the policy optimization but in the policy **evaluation** in order to learn a more accurate distributional critic. We also provide a more general motivation in the general reply.
> * **Extension of D4PG [1]** D4PG did not study the distribution over the gradient. Our work can be seen as a Sobolev extension of this work. Extending it required a new framework for multi-variate distribution RL akin to [2].
> * **Methodological contribution** Indeed Distributional RL and Sobolev training are known methods: but this work is the first to propose a combination of the two which required a novel design for the distributional critic and the distributional discrepancy measure (we use MMD on this new application of (Distributional Sobolev)).
> * **Limitation of deterministic policy gradient** Indeed, this is a limitation of our work but studying stochastic policy gradient would be out of scope for this paper. Our baselines are D4PG which is a distributional extension of DDPG and MAGE which is a Sobolev extension of DDPG. Furthermore, we work on value-based methods which exclude methods like PPO/A2C.
> * **Limitation of cVAE** We are quite confused by this sentence as a great part of the literature on model-based reinforcement involves variational inference and cVAEs [3] [4]
> * **Likelihood estimation issue** As we explained in this paragraph, modelling a distribution on a gradient-space isn't trivial as we want to preserve the Sobolev inductive bias. Since we want to model a gradient using the gradient of a neural network, we cannot easily put a probability distribution that would make density estimation possible. This rules out many methods for generative modelling such as normalizing flows or simply mixture of Gaussians as we explained. The reason is straightforward: even if one can model the likelihood of some value of a random variable, there is no obvious way to estimate the likelihood of some value of its gradient (with respect to a conditioning variable). Thus we must rely on a generative modelling scheme that learns only on the base of true samples and generated samples with not densities available.
> * **Meaning of the derivative of a random variable in Eq. 8** We did not invent this equation. This is the standard distributional deterministic policy gradient from D4PG paper [1]. We added a few words below the confusing equation and added an appendix section.
> * **Incomplete references** we corrected all the references.
> * **Supervised learning experiments** we argue theses experiments are useful to motivate the primary novely of this paper: distributional Sobolev training. We show that our implementation of this framework allows to effectively use gradient information to learn a more accurate distribution. We also showcases that a distributional estimation can have benefitial properties in the low data regime.
>
> We hope the revisions address your concerns and provide further clarity on the novelty and contributions of our method. Thank you once again for your time and constructive comments.
>
> [1] Distributed Distributional Deterministic Policy Gradients https://arxiv.org/abs/1804.08617
>
> [2] Distributional Reinforcement Learning for Multi-Dimensional Reward Functions https://openreview.net/pdf?id=u7oKU1iXTa9
>
> [3] MASTERING ATARI WITH DISCRETE WORLD MODELS https://arxiv.org/pdf/2010.02193
>
> [4] World Models https://arxiv.org/pdf/1803.10122

---

### Official Review · Reviewer_EgVG · 2024-10-31

**Soundness:** 1
**Presentation:** 1
**Contribution:** 2
**Rating:** 3
**Confidence:** 2

**Summary:**

This paper proposes “Distributional Sobolev RL” to extend distributional RL by modeling both the return and gradient distributions of the state-action value function. Using a one-step world model with a conditional Variational Autoencoder (cVAE) and Maximum Mean Discrepancy (MMD) for the Bellman operator, the approach is validated on toy tasks and Mujoco/Brax environments.

**Strengths:**

Learning the gradient in addition to the distribution is an interesting idea.

**Weaknesses:**

I failed to understand this paper and am quite confused about the proposed approach. We should check if it is a personal difficulty of me or if other reviewers also feel confused. If the latter, this indicates room for significant improvements in the paper.

I have listed my questions in the Questions section below.  These questions prevent me from understanding the proposed algorithm. In addition, Section 3.1 has few intuitive explanation. for example, why is eq (16) proposed? The paper states, "Depending on the applications Eq. 11 or 16 may make more sense ". This may need further clarification (when should we prefer 11 or 16?). Since I couldn’t follow Section 3.1, I was unable to understand Section 3.2 as well.

Overall, I find this paper lacks rigor and has limited (theoretical) justification. The definitions in section 3.1 are not rigorous enough.

The related work section is not organized well. The authors should put most related works here instead of in the introduction. Also, many prior works appear to be missed.

Minor issue: Figure 1 appear somewhat sloppy: the right side is cut off, and so $Z^{next}$ is incomplete.

**Questions:**

- The $f$ in eq (12) is not bold, which took me a while.
- In Line 178, how do you assume the action-gradient $x^\text{action}$ exists? First, I believe you need some differentiable conditions; second, if actions are discrete, how is the gradient defined in this case?
- Similarly, what if transition is discrete? Then is $\frac{\partial }{\partial s} x^{\text{return}}$ well defined?
- Similarly, is $\frac{\partial }{\partial a} R(s,a)$ well defined given that $a$ may be discrete?
- In Line 191, what does "gradient computations of the bootstrapped target" mean? Why do you assume it is differentiable?
- In eq (16), the notation $\nabla_s$ is unclear since there is no $s$ on the right. This is confusing.
- SImilarly, $\nabla_a$ is also unclear in eq (16). It is also in eq (11).
- Estimating the return distribution and the gradient seem orthogonal. Could you explain why you want to merge them in this paper?

---

> ### Author Response · Authors · 2024-11-23
>
> Thank you for taking the time to thoroughly review our work and provide valuable insights and constructive feedback.
>
> Here we try to address your questions and concerns.
>
> * **Equation 11 vs equation 16** we removed equation 16 as learning the state-action gradient is an additional but non-necessary improvement to the framework as it is and it brings confusion. In the removed paragraph, we wrote this to argue that as the state space is sometimes much larger in dimensionality than the action space, learning the state part of the gradient may bring more information from the world-model. Also, environments may exhibit different types of distribution whether on the state part or the action part of the return's gradient which could or could not be benefitial to model.
> * **Theoretical justification** we added a part in the approach section that motivates our framework (Section 3.1).
> * **Lack of rigor in the definitions** We added a section in the appendix for the derivation of the new operator defined in Eq 18. May we ask what parts you would like to see improved/clarified ?
> * **Existence of the gradient(s)** we were not clear enough about this. Our work only tackles continuous state-action spaces. It can be seen as a Sobolev extension of D4PG [2]. Furthermore, we assume this gradient exist and that we can calculate its value. It is an assumption that we then relax using the world-model similarly to [1]. We added a few words in the approach section (in red) to make this clearer.
> * **Meaning of *bootstrapped target*** we substantially modified this paragraph. It is hopefully clearer now. The bootstrapped target was related to re-using the current estimator for the distributional critic in the next state-action pair as in temporal-difference. We removed this phrasing as it brought more confusion. We also added a small derivation for the new operator in the appendix. We indeed assume it is differentiable. We made it clearer we start from this assumption and relax it later using the world-model similarly to [1].
> * **$\nabla_s$ and $\nabla_a$** the gradient is taken with respect to the application $a_0=a$. We added a sentence to clarify this.
>     \item Why learning the return distribution along the gradient ? First we clarify that we learn the \textbf{joint} distribution over the returns and their gradients. For the motivations we refer to the general reply above.
> * **Related works** We merge the related works with the introduction as was suggested by another reviewer. Furthermore, we mention several related works in the Section 3.3.
>
> We hope the revisions address your concerns and provide further clarity on the novelty and contributions of our method. Thank you once again for your time and constructive comments.
>
> [1] How to Learn a Useful Critic? Model-based Action-Gradient-Estimator Policy Optimization
>
> [2] Distributed Distributional Deterministic Policy Gradients https://arxiv.org/abs/1804.08617

---

> > ### Comment · Reviewer_EgVG · 2024-11-25
> >
> > Thank you to the authors for their response and for updating the manuscript. The new version is generally clearer than the previous one. However, the significant changes in the main section suggest that further refinements are needed and a second round of review would be beneficial. Additionally, the related works section remains incomplete, lacking many important references from distribution RL.

---

> > > ### Author Response · Authors · 2024-11-25
> > > **Related works modifications**
> > >
> > > We sincerely thank the reviewer for their valuable input and comments.
> > >
> > > In response, we have updated the *Related Works* section to include the most relevant seminal works in distributional reinforcement learning (RL) and multivariate distributional RL. Additionally, we clarified the connection to Sobolev training of neural networks and included a few sentences about model-based RL. **Given the expanded size of the *Related Works* section, we have relocated it to appear before the conclusion for better organization.**
> > >
> > > We would greatly appreciate it if the reviewer could kindly suggest any specific references that might still be missing. Such guidance would be tremendously helpful, and we thank the reviewer in advance for their time.

---

### Official Review · Reviewer_pyqn · 2024-11-04

**Soundness:** 3
**Presentation:** 2
**Contribution:** 2
**Rating:** 3
**Confidence:** 4

**Summary:**

The paper meticulously formulates a new model-based actor-critic method for distributional RL. The proposed method utilizes a conditional VAE as their world model, that learns to predict both the next state and reward, conditioned on the current state and action. This model, trained along with the policy and critic networks, allows computing the MMD loss for the critic, and introduces uncertainty in the action gradients (of the returns) along with the returns themselves. The authors elaborate the required mathematical background to explain their method and design choices, and show the advantage of the distributional approach and the usage of gradient information in a toy problem. Combined with RL, the proposed method is tested in five mujoco environments, showing a certain superiority over other approaches.

**Strengths:**

1. Mathematical background and notations: Comprehensive and helps to explain the proposed method. I am not sure which ones are novel though, if some of them, it is worth noting.

2. The extension to distributional RL is done elegantly, using a rather simple world model, which allows using other types of models.

3. Experiments seem extensive in terms of benchmarks, although see my notes in the weaknesses section.

**Weaknesses:**

### Experiments:
1.  My major concern with this work is the empirical performance of the proposed method; It performs very similar to DDPG with MMD (only minor improvements) and in most tested environments, vanilla DDPG performs the best. Considering that the toy problem does not involve RL, I think that the method is not justified enough. I suggest looking for other environments that show the value of such method, maybe ones that are more stochastic?

2. Lack of baselines: although the method is meant for the distributional setting, mean-based methods could be applied to the same environments, and there are many that should be compared, if not for direct comparison it helps for understanding the scale of performance improvement.

### Related Work:
This section seems very limited. Since the introduction describes some, I would move this section to the beginning of the paper, and maybe merge with the introduction section.

### Clarity:
For me, it is not clear enough what is novel in this method; The usage of the gradient info has been done, hence the method extends this approach to distributional RL?
The same applies for the mathematical analysis. I think it should be clearer -- either differ between existing and new/ add a statement of the paper's contribution.

### Minor Comments:
1. line 449: Deterministic Deep Policy Gradient -> Deep Deterministic Policy Gradient.

2. line 456: both variants of
DSDPG, using action gradient and DSDPG using state-action" -- rephrase

**Questions:**

1. What are the paper's contributions?

2. Is it possible to extend your method to support other actor-critic methods? (e.g., PPO, A2C, etc.)

3. In your experiments, it seems that you evaluate the learning curve, while eventually most methods perform in par after enough interaction. Have you tried limiting the data (to show better sample complexity)?

4. One of the advantages of using a model-based method, given an accurate enough model, is the ability to train without direct interaction (only from the world model) have you tried it?

---

> ### Author Response · Authors · 2024-11-23
>
> Thank you for taking the time to thoroughly review our work and provide valuable insights and constructive feedback.
>
> We will try to address your concerns and then answer your questions.
>
> # Concerns
>
> * **Performance**: We acknowledge the limited empirical performance of our method and are actively trying to improve it. We suspect the issue lies in the conditional VAE. As we rely on variational inference to infer gradients from true observations as explained in Appendix A.2 and A.3, the performance of the cVAE is critical. It must provide good reconstruction to *query* in the correct next-state and provide a accurate enough reward. However, just as critical is new sample generation. Indeed, since the gradients are inferred, the posterior (from the encoder) must be close to the prior while providing good reconstruction (which is related to good samples generation). This is a usual balance to find for Variational Autoencoder that has been particularly difficult to strike in this work.
> * **Environment stochasticity**: Regarding the *stochasticity* of the environments, we used N-step return (with N = 5) which makes the dynamics highly stochastic from the perspective of policy evaluation. Indeed, as explained in Section 2.2, N-step return uses the exploration policy for N steps which makes the mapping $(s, a) \rightarrow (s', r)$ non-deterministic even if it originally was. We further motivate this from D4PG [1] which showcased empirical improvements once making the environment stochastic this way.
> * **Baselines** About the baselines, we argue that only few are needed to motivate the empirical improvement. Indeed, DSDPG is not a model-based algorithm like the Dyna family [2]. We do not use the world-model for imagination but to provide a differentiable surrogate of true observations via inference akin to [3]. The primary reason is that we wanted our implementation of Distributional Sobolev Deterministic Policy Gradient to be as minimal as possible. Hence, we claim that only the following baselines are needed
>  - A scalar-based distributional method (MMD)
>  - A scalar-based deterministic method (DDPG)
>  - A gradient-based deterministic method (MAGE) [4]
>
> # Related works
> We merged this section with the introduction. Several related works are also discussed in Section 3.3.
>
> # Clarity
>
> **Novelty** We added a paragraph about the contributions in the introduction and in the general reply above. Indeed, our method can be seen as
>   - A distributional generalization of MAGE [4] that used Sobolev training in value-based RL **and/or**
>   - A Sobolev generalization of D4PG [1]
>
> It required developping new tools to model gradient as a random variable along random returns.
>
> **Mathematical section** We modified Section 3.2 heavily and tried to make it clearler what is new.
>
> # Questions
>
> * ***What are the paper's contributions ?*** We refer to the general reply and the added paragraph in the introduction.
> * ***Is it possible to extent your method to support other actor-critic methods ?*** probably but it would certainly not provide the same benefits as on value based methods such as DDPG/TD3/SAC. Indeed, we improve the critic by learning a distribution over gradient as the critic's action gradient is the one quantity that is used for policy improvement in this family of methods. The usual policy gradient and its PPO variant only use the critic through its scalar output (or the conditional mean in the case of distributional RL) which could be made more accurate by learning the gradient. Furthermore, as it seems other reviewers had similar question: our method is primarly designed for continuous action spaces. It would then rule out the discrete applications of PPO/A2C.
> * ***In your experiments, it seems that you evaluate the learning curve, while eventually most methods perform on par after enough interaction. Have you tried limiting the data (to show better sample complexity) ?*** we are not sure exactly what you propose. We implemented the usual training loop where the more training steps the more samples from the observations are collected and buffered. Would you mean reducing the number of actors (for exploration) or doing more learning steps per exploration step ? In which case, we didn't investigate in that direction.
> * ***One of the advantages of using a model-based method, given accurate enough model, is the ability to train without direct interaction (only from the world model) have you tried it ?*** we have not explored this possibility as our world-model is only used to infer gradient from true observations and not imagination as in Dyna [2]. Although this could be possible, this would be an orthogonal modification to the current framework. We stress again that our implementation was minimal and a proof of concept that DSDPG can work.
>
> We hope the revisions address your concerns and provide further clarity on the novelty and contributions of our method. Thank you once again for your time and constructive comments.

---

> ### Author Response · Authors · 2024-11-23
> **references**
>
> [1] Distributed Distributional Deterministic Policy Gradients https://arxiv.org/abs/1804.08617
>
> [2] Dyna, an integrated architecture for learning, planning, and reacting
>
> [3] Learning Continuous Control Policies by Stochastic Value Gradients https://arxiv.org/abs/1510.09142
>
> [4] How to Learn a Useful Critic? Model-based Action-Gradient-Estimator Policy Optimization

---

> > ### Comment · Reviewer_pyqn · 2024-11-27
> >
> > I thank the authors for their detailed response and the revisions of the paper.  While most of my concerns have been addressed, I am still not convinced that the experiments section is good enough. Regarding the performance: I understand the difficulties, but the suggested method needs to be justified. Maybe it is about finding the right setting; which types of environments/base algorithm. Another issue is the baselines, especially in the current performance, as I understand, this method can be paired with various algorithms, showing which "base" algorithm your proposed method effectively improves could be valuable.
> >
> > I think this paper has a good potential, but for my opinion, it requires more work and revisions, hence I leave my original score.

---

### Author Response · Authors · 2024-11-23
**General reply**

We thank the reviewers for taking the time to read our work and for their valuable insights and constructive feedbacks.

We observed some recurring points between the reviews, thus we first provide a general reply to those and answer in more details below the specific reviews. We also list the modifications we made to the submission:
# General questions
* **Discrete vs continuous action space**
It seems there was some confusion regarding the type of tasks our work tries to improve the baselines on. We work on __continuous state-action spaces__ and not discrete environments. Thus the gradients of the return and dynamics may exist although in practice they are most often unknown.
* **Motivations - why we model the distribution over action-gradient of the return distribution**
We do so primarily motivated by the empirical improvements of distributional RL and gradient learning value-based RL [3]. Furthermore, we argue that the action-gradient being a much more high dimensional quantity, learning a pointwise value might bring the same types of difficulties as for scalar return where the conditional expectation is difficult to capture as no observations match precisely this expectation and as there might be many modes in the distribution. We argue that the more dimensions, the more difficult the conditional expectation of the action-gradient might be to capture. Hence, we developed a framework to model the full distribution. Moreover, distributional policy evaluation is a *richer* task as it requires to model a richer object (the distribution over returns vs the expectation). This task being richer might also help learning better representations of the environment. Thus, we also argue that learning a distribution over action-gradient is a *richer* task than modeling the expected action-gradient.
* **Contributions**
  - Distributional Sobolev training: we proposed to incorporate gradient information while modelling conditional random variables which requires to model the gradient as a random variable as well. We instantiate our method by considering a pushforward generative model's outputs and gradients as random variable. We use moment matching to measure discrepancy between the target and predicted distributions via samples.
  - Conditional VAE - varitional inference for stochastic value gradients. As most often the gradient of reinforcement learning is undefined or unknown, we train a conditional Variational Autoencoder to **infer** the gradients from true observations (as opposed to imagination of new transitions in many model-based works [2] [3]). This can be seen as an extension of Stochastic Value Gradients [1].
* **Imagination vs inference** we do not use our world-model for imagination as is common in model-based RL [2] where the world-model is used to generate more training data. Instead, we use our cVAE world-model to reconstruct a differentiable surrogate of the next state and reward.
* **Performance** we acknowledge the limited performance benchmarks of our method. Our work was aimed as more of a proof of concept. We developed tools to model uncertainty over the gradient along the regular value of the return distribution thus extending distributional reinforcement learning. Furthermore, we demonstrated that, on a toy supervised learning task, our method was sound and effective.
# Modifications
Here we list the modifications we made to the submission which are in red (for the most part)
* Introduction: we added a part about our contribution. ~~We merged the related works section as was suggested.~~
* We enriched the related works section with seminal works in distributional and model-based RL.
* We added a requested theoretical motivation (Section 3.1). Where we motivate our approach of matching the gradient distributions. The proof is referenced in the Appendix.
* We clarified the Section 3.2 where we presented our Distributional Sobolev training framework. We also removed mentions of state-action gradient and only preserve the action gradient of the return distribution as it brought more confusion and was not necessary to motivate our framework overall. We modified the explanations regarding the new operator proposed in Eq. 18. We also added the derivation for this operator in the appendix.
* We removed mentions of the state-action gradient modeling and only kept the action-gradient modeling as we realized it brought more confusion and was not necessary in general. We only kept mentions of it in the experiment part but will remove them if necessary.
* We clarified the notion of derivative of a random variable with respect to a conditioning variable as first mentioned in Eq. 8. We also added details in the appendix.
# References
[1] Learning Continuous Control Policies by Stochastic Value Gradients https://arxiv.org/abs/1510.09142

[2] Dyna, an integrated architecture for learning, planning, and reacting

[3] How to Learn a Useful Critic? Model-based Action-Gradient-Estimator Policy Optimization

---

### Meta-Review · Area_Chair_JhT4 · 2024-12-08

**Metareview:**

The paper proposes a distributional Sobolev training method for distributional RL, which utilizes a differential world model represented by a conditional VAE to predict the future state and reward. In particular, beyond standard distributional RL that learns the distribution of returns, the proposed Sobolev technique aims to also learn the distribution of the gradient of the Q function, which may potentially improve the performance of policy gradient algorithms.

After the discussion between the authors and the reviewers, there still remain several significant issues that are not totally cleared. Especially, the numerical experiments and the baselines are not convincing enough in this paper, the motivation of the paper is not well justified, and the writing of the paper needs large improvements. In particular, as a direct combination of existing techniques, the contribution of the paper needs more careful justification.

Overall, we decide to reject this paper.

**Additional Comments On Reviewer Discussion:**

The main issues raised by the reviewers are about writing, motivation, contribution, experiments, and related works. After the rebuttal period, the authors have cleared up many of the concerns and has made many modifications in writing and related works. However, the concerns of the reviewers are not fully addressed. In particular in the experiments and the significance of the contribution.

---

### Decision · Program_Chairs · 2025-01-22

Reject